# Outcome-Based Online Reinforcement Learning: Algorithms and Fundamental Limits

**Fan Chen**[*]
fanchen@mit.edu

**Zeyu Jia**[*]
zyjia@mit.edu

**Alexander Rakhlin**[*]
rakhlin@mit.edu

**Tengyang Xie**[†]
tx@cs.wisc.edu

## Abstract

Reinforcement learning with outcome-based feedback faces a fundamental challenge: when rewards are only observed at trajectory endpoints, how do we assign credit to the right actions? This paper provides the first comprehensive analysis of this problem in online RL with general function approximation. We develop a provably sample-efficient algorithm achieving $\widetilde{O}(C_{\mathrm{cov}} H^3/\varepsilon^2)$ sample complexity, where $C_{\mathrm{cov}}$ is the coverability coefficient of the underlying MDP. By leveraging general function approximation, our approach works effectively in large or infinite state spaces where tabular methods fail, requiring only that value functions and reward functions can be represented by appropriate function classes. Our results also characterize when outcome-based feedback is statistically separated from per-step rewards, revealing an unavoidable exponential separation for certain MDPs. For deterministic MDPs, we show how to eliminate the completeness assumption, dramatically simplifying the algorithm. We further extend our approach to preference-based feedback settings, proving that equivalent statistical efficiency can be achieved even under more limited information. Together, these results constitute a theoretical foundation for understanding the statistical properties of outcome-based reinforcement learning.

## 1 Introduction

Reinforcement learning with outcome-based feedback is a fundamental paradigm where agents receive rewards only at the end of complete trajectories rather than at individual steps. This feedback model naturally arises in many applications, from large language model training (Ouyang et al., 2022; Bai et al., 2022; Jaech et al., 2024), where human preferences are provided for entire outputs rather than individual tokens, to clinical trials, where patient outcomes are only observable after a complete treatment regimen. Despite the prevalence of such settings, the statistical implications of outcome-based feedback for online exploration remain poorly understood.

In traditional reinforcement learning (Sutton et al., 1998), agents observe rewards immediately after each action, providing a granular signal that directly links actions to their consequences. In contrast, outcome-based feedback presents a fundamental challenge: when rewards are only observed at the trajectory level,determining which specific actions contributed to the final outcome becomes significantly more difficult. This credit assignment problem is particularly acute in sequential decision-making tasks with long horizons, where many different action combinations could lead to the observed outcome.

While recent work (Jia et al., 2025) has shown that outcome-based feedback is sufficient for offline reinforcement learning under certain conditions, the feasibility of efficient online exploration with only trajectory-level feedback remains an open question. Online learning—where an agent actively explores to gather new data—is essential for adaptive systems that must learn in dynamic environments without pre-collected datasets. This leads to our central question:

*When is online exploration with outcome-based reward statistically tractable?*

39th Conference on Neural Information Processing Systems (NeurIPS 2025).

This question has been studied in the setting where the reward function is assumed to be well-structured (Efroni et al., 2021; Pacchiano et al., 2021; Chatterji et al., 2021; Cassel et al., 2024; Lancewicki and Mansour, 2025), with a primary focus on the *linear* reward functions. Similar reliance on the well-behaved reward structure[3] also appears in the recent work on Reinforcement Learning from Human Feedback (RLHF) (Chen et al., 2022b,a; Wu and Sun, 2023; Wang et al., 2023), where only *preference* feedback is available. However, well-behaved reward structure is dedicated and might fail to capture many real-world scenarios with general function approximation. In this paper, we address this question by providing a comprehensive theoretical analysis of outcome-based online reinforcement learning with general function approximation. We investigate when efficient exploration is possible with only trajectory-level feedback and characterize the fundamental statistical limits of learning in this setting. Our main results are as follows:

(1) We present a model-free algorithm for outcome-based online RL with general function approximation (Algorithm 1) that relies solely on trajectory-level reward feedback rather than per-step feedback. Our algorithm achieves a complexity bound of $\widetilde{O}(C_{\mathrm{cov}}H^3/\varepsilon^2)$ under standard realizability and completeness assumptions, where $C_{\mathrm{cov}}$ is the coverability coefficient that measures an intrinsic complexity of the underlying MDP. This bound applies in the general function approximation setting where state spaces may be large or infinite, requiring only that value functions can be represented by an appropriate function class with bounded statistical complexity.

(2) For the special case of deterministic MDPs, we present a simpler algorithm based on Bellman residual minimization (Algorithm 2) that achieves similar theoretical guarantees with improved computational efficiency.

(3) As extension, we generalize our approach to preference-based reinforcement learning (Section 4), where feedback comes in the form of binary preferences between trajectory pairs under the Bradley-Terry-Luce model. This extension bridges the gap to practical reinforcement learning from human feedback (RLHF) scenarios, where even outcome reward feedback is rare.

(4) We also identify a fundamental separation between outcome-based and per-step feedback (Section 5). Specifically, there exists a MDP with known transition dynamics and horizon $H = 2$, and the reward being a $d$-dimensional generalized linear function, while in this problem $e^{\Omega(d)}$ samples are necessary to learn a near-optimal policy with only outcome reward. However, such a problem is known to be *easy* with per-step reward feedback, in the sense that existing algorithms can return an $\varepsilon$-optimal within $\widetilde{O}(d^2/\varepsilon^2)$ rounds with per-step feedback. This separation demonstrates that delicate analysis based on well-behaved reward structure can fail catastrophically when only outcome reward feedback is available.

Our results provide a theoretical foundation for understanding when outcome-based exploration is tractable and when it presents insurmountable statistical barriers. By characterizing these fundamental limits, we offer guidance for the development of efficient algorithms for learning from trajectory-level feedback in online settings and highlight the precise conditions under which outcome-based feedback is statistically equivalent to per-step feedback.

## 2   Preliminaries

**Markov Decision Process.**   An MDP $M$ is specified by a tuple $(\mathcal{S}, \mathcal{A}, \mathbb{T}, \rho, R, H)$, with state space $\mathcal{S}$, action space $\mathcal{A}$, horizon $H$, transition kernel $\mathbb{T} = (\mathbb{T}_h : \mathcal{S} \times \mathcal{A} \to \Delta(\mathcal{S}))_{h=1}^{H-1}$, is initial state distribution $\rho \in \Delta(\mathcal{S})$, and the mean reward function $R = (R_h : \mathcal{S} \times \mathcal{A} \to [0,1])_{h=1}^{H}$. At the start of each episode, the environment randomly draws an initial state $s_1 \sim \rho$, and then at each step $h \in [H]$, after the agent takes action $a_h$, the environment generates the next state $s_{h+1} \sim \mathbb{T}(\cdot|s_h, a_h)$. The episode terminates immediately after $a_H$ is taken, and, for notational simplicity, we denote $s_{H+1}$ to be the deterministic terminal state. We denote $\tau = (s_1, a_1, \cdots, s_H, a_H)$ to be the trajectory, and throughout this paper we always assume the reward function is normalized, i.e., $R(\tau) := \sum_{h=1}^{H} R_h(s_h, a_h) \in [0,1]$ almost surely.

In addition to the states, the learner may also observe the reward feedback after the episode terminates. In the *process reward* feedback setting, the learner receives a random reward vector $(r_1, \cdots, r_H) \in$

---

[3]More specifically, most of the recent work either assume the reward class is linear or admits low eluder dimension (as a function of the *trajectory*).

$[0,1]^H$ such that $\mathbb{E}[r_h|\tau] = R_h(s_h, a_h)$ for each $h \in [H]$. In the *outcome reward* setting, the learner only receives a single reward value $r \in [H]$ such that $\mathbb{E}[r|\tau] = \sum_{h=1}^H R_h(s_h, a_h)$.

**Policies, value functions, and the Bellman operator.** A (randomized) policy $\pi$ is specified as $\{\pi_h : \mathcal{S} \to \Delta(\mathcal{A})\}$, and it induces a distribution $\mathbb{P}^\pi$ of trajectory $\tau = (s_1, a_1, \cdots, s_H, a_H)$ by $s_1 \sim \rho$, and for each $h \in [H]$, $a_h \sim \pi_h(s_h)$, $s_{h+1} \sim \mathbb{T}_h(s_h, a_h)$. We let $\mathbb{E}^\pi[\cdot]$ to be the corresponding expectation.

The expected cumulative reward of a policy $\pi$ is given by $J(\pi) := \mathbb{E}^\pi\left[\sum_{h=1}^H R_h(s_h, a_h)\right]$. The value function and $Q$-function of $\pi$ is defined as

$$V_h^\pi(s) := \mathbb{E}^\pi\left[\sum_{\ell=h}^H R_\ell(s_\ell, a_\ell)\,\middle|\, s_h = s\right], \qquad Q_h^\pi(s, a) := \mathbb{E}^\pi\left[\sum_{\ell=h}^H R_\ell(s_\ell, a_\ell)\,\middle|\, s_h = s, a_h = a\right].$$

Let $\pi^\star$ denote an optimal policy (i.e., $\pi^\star \in \operatorname{argmax}_\pi J(\pi)$), and let $V^\star$ and $Q^\star$ be the corresponding value function and $Q$-function. It is well-known that $(V^\star, Q^\star)$ satisfies the following Bellman equation for each $s \in \mathcal{S}, a \in \mathcal{A}, h \in [H]$:

$$V_h^\star(s) = \max_{a \in \mathcal{A}} Q_h^\star(s, a), \qquad Q_h^\star(s, a) = R_h(s, a) + \mathbb{E}_{s' \sim \mathbb{T}_h(\cdot|s,a)} V_{h+1}^\star(s'), \tag{1}$$

with the convention that $V_{H+1}^\star = 0$. Therefore, we define the Bellman operator $\mathcal{T}_h$ as follows: for any $f : \mathcal{S} \times \mathcal{A} \to \mathbb{R}$, $\mathcal{T}_h f$ is defined as

$$[\mathcal{T}_h f](s, a) := R_h(s, a) + \mathbb{E}_{s' \sim \mathbb{T}_h(\cdot|s,a)} \max_{a' \in \mathcal{A}} f(s', a'). \tag{2}$$

Then, it is straightforward to verify that the Bellman equation reduces to $Q_h^\star = \mathcal{T}_h Q_{h+1}^\star$ for $h \in [H]$.

**Complexity measure of the MDP.** Coverability is a natural notion for measuring the difficulty of learning in the underlying MDP (Xie et al., 2022).

**Definition 1** (Coverability). *For a given MDP $M$ and a policy class $\Pi$, the coverability $C_{\mathrm{cov}}$ is defined as*

$$C_{\mathrm{cov}}(\Pi; M) := \min_{\mu_1, \cdots, \mu_H \in \Delta(\mathcal{S} \times \mathcal{A})} \max_{h \in [H], \pi \in \Pi} \left\|\frac{d_h^\pi}{\mu_h}\right\|_\infty,$$

*where $\left\|\frac{d_h^\pi}{\mu_h}\right\|_\infty := \max_{s \in \mathcal{A}, a \in \mathcal{A}} \frac{d_h^\pi(s,a)}{\mu_h(s,a)}$.*

The coverability coefficient of an MDP is an inherent measure of the diversity of the state-action distributions. Our main upper bounds scale with the coverability of the underlying MDP $M^\star$, and in this case we abbreviate $C_{\mathrm{cov}}(\Pi) := C_{\mathrm{cov}}(\Pi; M^\star)$ for succinctness.

**Function approximation.** In this paper, we work with (model-free) function approximation, where the learner have access to a *value function class* $\mathcal{F} = \mathcal{F}_1 \times \cdots \times \mathcal{F}_H$ and a *reward function class* $\mathcal{R} = \mathcal{R}_1 \times \cdots \times \mathcal{R}_H$ with each $\mathcal{F}_h, \mathcal{R}_h \subseteq (\mathcal{S} \times \mathcal{A} \to [0, 1])$.

The function class $\mathcal{F}$ and $\mathcal{R}$ consist of candidate functions to approximate $Q^\star$ and the ground-truth reward function $R^\star$.[4] In the literature of RL with general function approximation, it is typically assumed that the function classes are *realizable*, i.e., $Q^\star \in \mathcal{F}$ and $R^\star \in \mathcal{R}$. In this paper, we adopt the following relaxed realizability condition with a fixed approximation error $\varepsilon_{\mathrm{app}} \geq 0$.

**Assumption 1** (Realizability). *There exists $Q^\sharp \in \mathcal{F}$ and $R^\sharp \in \mathcal{R}$ such that $\max_{h \in [H]} \|Q_h^\sharp - Q_h^\star\|_\infty \leq \varepsilon_{\mathrm{app}}$, $\max_{h \in [H]} \|R_h^\sharp - R_h^\star\|_\infty \leq \varepsilon_{\mathrm{app}}$.*

For each value function $f \in \mathcal{F}$, it induces a greedy policy $\pi_f$ given by $\pi_{f,h}(s) := \operatorname{argmax}_{a \in \mathcal{A}} f(s, a)$. Therefore, the value function class $\mathcal{F}$ induces a policy class $\Pi_\mathcal{F} := \{\pi_f : f \in \mathcal{F}\}$, and we take our policy class $\Pi = \Pi_\mathcal{F}$ for the remaining part of this paper.

The complexity of the function class is measured by the covering number.

**Definition 2** (Covering number). *For a function class $\mathcal{H} \subseteq (\mathcal{X} \to \mathbb{R})$ and parameter $\alpha \geq 0$, an $\alpha$-covering of $\mathcal{H}$ (with respect to the sup norm) is a subset $\mathcal{H}' \subseteq \mathcal{H}$ such that for any $f \in \mathcal{H}$, there exists $f' \in \mathcal{H}'$ with $\sup_{x \in \mathcal{X}} |f(x) - f'(x)| \leq \alpha$. We define the $\alpha$-covering number of $\mathcal{H}$ as $N(\mathcal{H}, \alpha) := \min\{|\mathcal{H}'| : \mathcal{H}' \text{ is a } \alpha\text{-covering of } \mathcal{H}\}$.*

---

[4]In the following, we always write $R^\star$ for the true reward function to avoid confusion.

**Bellman operator.** A reward function $R = (R_1, \cdots, R_H) \in \mathcal{R}$ induces a Bellman operator as

$$[\mathcal{T}_{R,h} f](s, a) := R_h(s, a) + \mathbb{E}_{s' \sim \mathbb{T}_h(\cdot|s,a)} \max_{a' \in \mathcal{A}} f_{h+1}(s', a'), \qquad \forall f = (f_1, \cdots, f_H) \in \mathcal{F},$$

where we also adopt the notation $f_{H+1} = 0$ for any $f \in \mathcal{F}$. Most literature on RL with general function approximation also makes use of a richer comparator function class $\mathcal{G} = \mathcal{G}_1 \times \cdots \times \mathcal{G}_H$ that satisfies the following *Bellman completeness* (Jin et al., 2021a; Xie et al., 2021, 2022, etc.).

**Assumption 2** (Bellman completeness). *For each $h \in [H]$, $\mathcal{F}_h \subseteq \mathcal{G}_h$. For any $f \in \mathcal{F}$ and $R \in \mathcal{R}$, it holds $\inf_{g_h \in \mathcal{G}_h} \|\mathcal{T}_{R,h} f - g_h\|_\infty \leq \varepsilon_{\text{app}}$ for $h \in [H]$.*

**Miscellaneous notation.** For any $p \in [0, 1]$, we define $\text{Bern}(p)$ to be the Bernoulli distribution with $\mathbb{P}(X = 1) = p$. For functions $f$ and $g \geq 0$, we use $f = O(g)$ to denote that there exists a universal constant $C$ such that $f \leq C \cdot g$.

# 3 Sample-Efficient Online RL with Outcome Reward

In this section, we present a model-free RL algorithm with outcome reward, which achieves sample complexity guarantee scaling with the coverability coefficient and the log-covering number of the function classes.

## 3.1 Main Result

We present Algorithm 1, which is based on the principle of optimism. For simplicity of presentation, we assume that the initial state $s_1$ is fixed.

The crux of the proposed algorithm is a new method for performing Fitted-Q Iteration with only outcome reward, in contrast to most existing RL algorithms (with general function approximation) that make use the process reward $(r_1, \cdots, r_H)$ to fit the Q-function for each step (Du et al., 2021; Jin et al., 2021a, etc.). A natural first idea is to fit, given a dataset $\mathcal{D} = \{(\tau, r)\}$ consisting of previously observed (trajectory, outcome reward) pairs, a reward model from the reward function class $\mathcal{R}$ by optimizing the following reward model loss:

$$\mathcal{L}_{\mathcal{D}}^{\text{RM}}(R) := \sum_{(\tau, r) \in \mathcal{D}} \left( \sum_{h=1}^{H} R_h(s_h, a_h) - r \right)^2. \tag{3}$$

As discussed below in Remark 1, directly fitting an estimated reward model based on $\mathcal{L}_{\mathcal{D}}^{\text{RM}}$ can lead to bad performance. Instead, our algorithm jointly optimizes over the value functions and reward models, as detailed below.

For any proxy reward model $R \in \mathcal{R}$ and a value function $f \in \mathcal{F}$, we define the Bellman error at step $h \in [H]$ as

$$\mathcal{E}_{\mathcal{D}, h}(f_h, f_{h+1}; R) := \sum_{(\tau, r) \in \mathcal{D}} \left( f_h(s_h, a_h) - R_h(s_h, a_h) - \max_{a'} f_{h+1}(s_{h+1}, a') \right)^2, \tag{4}$$

a measure of violation of the Bellman equation (1) with the proxy reward model $R$. Then, we introduce the Bellman loss defined as

$$\mathcal{L}_{\mathcal{D}}^{\text{BE}}(f; R) := \sum_{h=1}^{H} \mathcal{E}_{\mathcal{D}, h}(f_h, f_{h+1}; R) - \inf_{g \in \mathcal{G}} \sum_{h=1}^{H} \mathcal{E}_{\mathcal{D}, h}(g_h, f_{h+1}; R), \tag{5}$$

where we subtract the infimum of $g \in \mathcal{G}$ over the helper function class $\mathcal{G}$, a common approach to overcoming the double-sampling problem (Antos et al., 2008; Zanette et al., 2020; Jin et al., 2021a; Liu et al., 2023b).

**Algorithm.** First fix an arbitrary policy $\pi_{\text{ref}}$ (can be the policy which takes an arbitrary action $a_0$ at al states). The proposed algorithm takes in a value function class $\mathcal{F}$, a reward function class $\mathcal{R}$ and a comparator function class $\mathcal{G}$, and performs the following two steps for each iteration $t = 1, 2, \cdots, T$:

1. (Optimism) Compute optimistic estimates of $(Q^\star, R^\star)$ through solving the following joint maximization problem with the dataset $\mathcal{D}$ consisting of all previously observed (trajectory, outcome reward) pairs:

$$(f^{(t)}, R^{(t)}) = \max_{f \in \mathcal{F}, R \in \mathcal{R}} \lambda f_1(s_1) - \mathcal{L}_{\mathcal{D}}^{\text{BE}}(f; R) - \mathcal{L}_{\mathcal{D}}^{\text{RM}}(R), \tag{6}$$

---

**Algorithm 1** Outcome-Based Exploration with Optimism

---

**input:** Q-function class $\mathcal{F}$, reward function class $\mathcal{R}$, comparator class $\mathcal{G}$, parameter $\lambda > 0$, reference policy $\pi_{\mathrm{ref}}$.

**initialize:** $\mathcal{D} \leftarrow \emptyset$.

1: **for** $t = 1, 2, \ldots, T$ **do**
2:     Compute the optimistic estimates:

$$(f^{(t)}, R^{(t)}) = \max_{f \in \mathcal{F}, R \in \mathcal{R}} \lambda f_1(s_1) - \mathcal{L}_{\mathcal{D}}^{\mathsf{BE}}(f; R) - \mathcal{L}_{\mathcal{D}}^{\mathsf{RM}}(R).$$

3:     Select policy $\pi^{(t)} \leftarrow \pi_{f^{(t)}}$.
4:     **for** $h = 1, 2, \cdots, H$ **do**
5:         Execute $\pi^{(t)} \circ_h \pi_{\mathrm{ref}}$ for one episode and obtain $(\tau^{(t,h)}, r^{(t,h)})$
6:         Update dataset: $\mathcal{D} \leftarrow \mathcal{D} \cup \{(\tau^{(t,h)}, r^{(t,h)})\}$.
7:     **end for**
8: **end for**
9: Output $\widehat{\pi} = \mathrm{Unif}(\pi^{(1:T)})$.

---

where for any $f \in \mathcal{F}$ we denote $f_1(s_1) := \max_{a \in \mathcal{A}} f_1(s_1, a)$ to be the value of $f$ at the initial state. Therefore, the optimization problem (6) enforces optimism by balancing the estimated value $f_1(s_1)$ and the estimation error $\mathcal{L}_{\mathcal{D}}^{\mathsf{BE}}(f; R) + \mathcal{L}_{\mathcal{D}}^{\mathsf{RM}}(R)$ though a hyper-parameter $\lambda \geq 0$.

2. (Data collection) Based on the optimism estimate $f^{(t)}$, the algorithm selects $\pi^{(t)} := \pi_{f^{(t)}}$. To collect data, the algorithm then executes the exploration policies $\pi \circ_h \pi_{\mathrm{ref}}$ for each $h \in [H]$, where for any policy $\pi$ and $\pi_{\mathrm{ref}}$, we let $\pi \circ_h \pi_{\mathrm{ref}}$ be the policy that executes $\pi$ for the first $h$ steps, and then executes $\pi_{\mathrm{ref}}$ starting at the $(h+1)$-th step.

**Theoretical analysis.** For Algorithm 1, we provide the following sample complexity guarantee, which scales with the coverability $C_{\mathrm{cov}} = C_{\mathrm{cov}}(\Pi_{\mathcal{F}})$, where $\Pi_{\mathcal{F}} = \{\pi_f : f \in \mathcal{F}\}$ is the policy class induced by $\mathcal{F}$. To simplify the presentation, we denote $\log N_T := \inf_{\alpha \geq 0} (\log N(\alpha) + T\alpha)$, where $N(\alpha)$ is defined as

$$N(\alpha) := \max_{h \in [H]} \{N(\mathcal{F}_h, \alpha), N(\mathcal{R}_h, \alpha), N(\mathcal{G}_h, \alpha)\}.$$

With the function classes being parametric, it is clear that $\log N_T \leq O(d \log(T))$.

**Theorem 1.** *Let $\delta \in (0, 1)$. Suppose that Assumption 1 and Assumption 2 hold, and the parameters are chosen as*

$$\lambda = c_0 \max \left\{ \frac{H \log(N_{TH^2}/\delta)}{\varepsilon}, TH\varepsilon_{\mathrm{app}} \right\}, \qquad T \geq c_1 \frac{C_{\mathrm{cov}} H^2 \log(T)}{\varepsilon^2} \cdot \log(N_{TH^2}/\delta), \quad (7)$$

*where $c_0, c_1 > 0$ are absolute constants. Then with probability at least $1 - \delta$, the output policy $\widehat{\pi}$ of Algorithm 1 satisfies $V^{\star}(s_1) - V^{\widehat{\pi}}(s_1) \leq \varepsilon + O(C_{\mathrm{cov}} H^2 \log(T) \cdot \varepsilon_{\mathrm{app}})$.*

The proof of Theorem 1 is deferred to Appendix C. Particularly, we note that when the function classes satisfy $\log N_T \leq \widetilde{O}(d)$ and $\varepsilon_{\mathrm{app}} = 0$, Algorithm 1 outputs an $\varepsilon$-optimal policy with sample complexity

$$TH \leq \widetilde{O}\left(\frac{C_{\mathrm{cov}} d H^3}{\varepsilon^2}\right).$$

Notably, the coverability $C_{\mathrm{cov}}$ measures the inherent complexity of the underlying MDP $M^{\star}$ (Xie et al., 2022) and it is independent of the reward function class. As our result only depends on the coverability $C_{\mathrm{cov}}$ and the statistical complexity of the function classes, it does *not* rely on the structure of reward functions, while previous works assume the reward functions are either linear (Efroni et al., 2021; Cassel et al., 2024) or admit low *trajectory* eluder dimension (Chen et al., 2022b,a).

**Remark 1.** *In Algorithm 1, the reward functions $R^{(t)}$ and Q-functions $f^{(t)}$ are jointly optimized (see Eq. (6)). A natural question is whether these can be optimized separately—i.e., first learning a fitted reward model and then applying optimism to the Q-functions based on the learned reward model. We*

---

**Algorithm 2** Outcome-Based Exploration with Optimism for Determinsitic MDP

---

**input:** Function class $\mathcal{F}$, parameter $\lambda > 0$.
**initialize:** $\mathcal{D} \leftarrow \emptyset$, initial estimate $f^{(1)} \in \mathcal{F}$.

1: **for** $t = 1, 2, \ldots, T$ **do**
2:     Receive $s^{(t)}$ and compute the optimistic estimates:

$$f^{(t)} = \max_{f \in \mathcal{F}} \lambda f_1(s_1^{(t)}) - \mathcal{L}_{\mathcal{D}}^{\mathsf{BR}}(f).$$

3:     Select policy $\pi^{(t)} \leftarrow \pi_{f^{(t)}}$.
4:     Execute $\pi^{(t)}$ to obtain a trajectory $\tau^{(t)} = (s_1^{(t)}, a_1^{(t)}, \ldots, s_H^{(t)}, a_H^{(t)})$ with outcome reward $r^{(t)}$.
5:     Update dataset: $\mathcal{D} \leftarrow \mathcal{D} \cup \{(\tau^{(t)}, r^{(t)})\}$.
6: **end for**
7: Output $\widehat{\pi} = \mathrm{Unif}(\pi^{(1:T)})$.

---

*show that this decoupled approach can lead to failures: due to reward model mismatch, the algorithm may become 'trapped' in regions where the exploratory policy fails to gather informative data. As a result, the sample complexity can become infinite in the worst case. See Section F.1 in the appendix for details.*

### 3.2 A Simpler Algorithm for Deterministic MDPs

A disadvantage of Algorithm 1 is that it requires solving a max-min optimization problem (6), as the Bellman loss $\mathcal{L}_{\mathcal{D}}^{\mathsf{BE}}$ involves a minimization problem over $\mathcal{G}$. While such computationally inefficient optimization problems are the common subroutines of existing function approximation RL algorithms (Jin et al., 2021a; Foster et al., 2021, 2022; Chen et al., 2022a, etc.), it turns out that Algorithm 1 can be significantly simplified when the transition dynamics in underlying MDP are deterministic.

**Assumption 3.** *The transition kernel $\mathbb{T}$ is deterministic, i.e., for any $h \in [H]$ and $s_h \in \mathcal{S}, a_h \in \mathcal{A}$, there is a unique state $s_{h+1} \in \mathcal{S}$ such that $\mathbb{T}_h(s_{h+1} \mid s_h, a_h) = 1$.*

Note that in this setting, the initial state $s_1$ and the outcome reward $r$ can still be random. This setting is also referred to as *Deterministic Contextual MDP* in Xie et al. (2024).

**Value difference as reward model.** A key observation is that, when the underlying MDP $M^\star$ is deterministic, the Bellman equation (1) trivially reduces to the following equality

$$Q_h^\star(s_h, a_h) = R_h^\star(s_h, a_h) + V_{h+1}^\star(s_{h+1}),$$

which holds almost surely. Hence, for any trajectory $\tau$, it holds that

$$R^\star(\tau) = \sum_{h=1}^{H} R_h^\star(s_h, a_h) = \sum_{h=1}^{H} \left[ Q_h^\star(s_h, a_h) - V_{h+1}^\star(s_{h+1}) \right].$$

Therefore, any value function $f \in \mathcal{F}$ induces an outcome reward model $R^f : (\mathcal{S} \times \mathcal{A})^H \to \mathbb{R}$ defined as

$$R^f(\tau) = \sum_{h=1}^{H} [f_h(s_h, a_h) - f_{h+1}(s_{h+1})],$$

where we adopt the notation $f_h(s) := \max_{a \in \mathcal{A}} f_h(s, a)$ for $h \in [H]$. This observation motivates the following Bellman Residual loss:

$$\mathcal{L}_{\mathcal{D}}^{\mathsf{BR}}(f) := \sum_{(\tau, r) \in \mathcal{D}} \left( \sum_{h=1}^{H} [f_h(s_h, a_h) - f_{h+1}(s_{h+1})] - r \right)^2, \tag{8}$$

where $\mathcal{D} = \{(\tau, r)\}$ is any dataset consisting of (trajectory, outcome reward) pairs.

**Bellman Residual Minimization (BRM) with Optimism.** For deterministic MDP, we propose Algorithm 2 as a simplification of our main algorithm. Similar to Algorithm 1, the proposed algorithm takes in the value function class $\mathcal{F}$ and alternates between the following two steps for each round $t = 1, 2, \cdots, T$:

1. (Optimism) Compute optimistic estimates of $Q^\star$ through solving the following maximization problem with the dataset $\mathcal{D}$ consisting of all previously observed (trajectory, outcome reward) pairs:

$$f^{(t)} = \max_{f \in \mathcal{F}} \lambda f_1(s_1) - \mathcal{L}_{\mathcal{D}}^{\mathsf{BR}}(f), \tag{9}$$

   enforcing optimism by balancing the estimated value $f_1(s_1)$ and the Bellman residual loss $\mathcal{L}_{\mathcal{D}}^{\mathsf{BR}}(f)$.

2. (Data collection) Based on the optimistic estimate $f^{(t)}$, selects $\pi^{(t)} := \pi_{f^{(t)}}$ and collect a trajectory $\tau^{(t)} = (s_1^{(t)}, a_1^{(t)}, \ldots, s_H^{(t)}, a_H^{(t)})$ with outcome reward $r^{(t)}$.

Compared to Algorithm 1, Algorithm 2 has the several advantages. First, it does not rely on the reward function class $\mathcal{R}$ and the comparator function class $\mathcal{G}$, and the Bellman residual loss $\mathcal{L}_{\mathcal{D}}^{\mathsf{BR}}$ is much simpler than the Bellman loss $\mathcal{L}_{\mathcal{D}}^{\mathsf{BE}}$, thanks to the deterministic nature of the underlying MDP. Therefore, Algorithm 2 is more amenable to computationally efficient implementation, as it replaces the max-min optimization problem (6) in Algorithm 1 with a much simpler maximization problem (9). Further, for every round $t$, the algorithm only needs to collect one episode from the greedy policy $\pi^{(t)}$.

**Theoretical analysis.** We present the upper bound of Algorithm 2 in terms of the following notion of coverability,

$$C'_{\mathrm{cov}}(\Pi) := \mathbb{E}_{s_1 \sim \rho} C_{\mathrm{cov}}(\Pi; M_{s_1}^\star),$$

where $M^\star$ is the underlying MDP, $M_{s_1}^\star$ is the MDP with deterministic initial state $s_1$ and the same transition dynamics as $M^\star$, and $\Pi = \Pi_{\mathcal{F}}$ is the policy class induced by $\mathcal{F}$. In general, $C'_{\mathrm{cov}}(\Pi)$ is always an upper bound on the coverability $C_{\mathrm{cov}}(\Pi)$, and the guarantee of Algorithm 2 scales with $C'_{\mathrm{cov}}(\Pi)$ as it avoids the layer-wise exploration strategy of Algorithm 1. We also denote

$$\log N_{\mathcal{F},T} := \inf_{\alpha \geq 0} \left( \max_{h \in [H]} N(\mathcal{F}_h, \alpha) + T\alpha \right).$$

**Theorem 2.** *Let $\delta \in (0,1)$. Suppose that Assumption 1 holds, and the parameters are chosen as*

$$\lambda = c_0 \max \left\{ \frac{H^3 \log(N_{\mathcal{F},T}/\delta)}{\varepsilon}, T\varepsilon_{\mathrm{app}} \right\}, \quad T \geq c_1 \frac{C'_{\mathrm{cov}}(\Pi) H^4 \log(T)}{\varepsilon^2} \cdot \log(N_{\mathcal{F},T}/\delta), \tag{10}$$

*where $c_0, c_1 > 0$ are absolute constants. Then with probability at least $1 - \delta$, Algorithm 2 achieves*

$$\frac{1}{T} \sum_{t=1}^{T} \left( V^\star(s_1^{(t)}) - V^{\pi^{(t)}}(s_1^{(t)}) \right) \leq \varepsilon + O(C'_{\mathrm{cov}}(\Pi) H \log(T) \cdot \varepsilon_{\mathrm{app}}).$$

The above upper bound provides the PAC guarantee through the standard online-to-batch conversion, and its proof is deferred to Appendix D. It is worth noting that Theorem 2 only relies on realizability assumption on the Q-function class $\mathcal{F}$, significantly relaxing the assumptions of realizablity (Assumption 1) and completeness (Assumption 2) in Theorem 1. As a remark, we note that Theorem 2 implies that Algorithm 2 in fact achieves a regret bound of order $\sqrt{T}$.[5]

## 4 Preference-based Reinforcement Learning

The goal of preference-based learning is to find a near-optimal policy only through interacting with the environment that provides *preference feedback*. As an extension of our results presented in Section 3, in this section we present a similar algorithm for preference-based RL with the same sample complexity guarantee.

---

[5]The (expected) *regret* of the algorithm can be defined as $\mathbf{Reg}(T) := \mathbb{E}\left[ \sum_{t=1}^{T} (V^\star(s_1^{(t)}) - V^{\pi^{(t)}}(s_1^{(t)})) \right]$.

**Preference-based learning in MDP.** In preference-based RL, the interaction protocol of the learner with the environment is specified as follows. For each round $t = 1, 2, \cdots$,

- The learner selects policy $\pi^{(t,+)}$ and $\pi^{(t,-)}$.

- The learner receives trajectories $\tau^{(t,+)} \sim \pi^{(t,+)}$, $\tau^{(t,-)} \sim \pi^{(t,-)}$, and *preference feedback* $y^{(t)} \sim \text{Bern}(\mathsf{C}(\tau^{(t,+)}, \tau^{(t,-)}))$, where $\mathsf{C}$ is a comparison function.

Intuitively, for any trajectory pair $(\tau^+, \tau^-)$, the comparison function $\mathsf{C}(\tau^+, \tau^-) = \mathbb{P}(\tau^+ \succ \tau^-)$ measures the probability that $\tau^+$ is more preferred. In this paper, we mainly focus on the Bradley-Terry-Luce (BTR) model (Bradley and Terry, 1952), which is widely used on RLHF literature. We expect that our algorithm and analysis techniques apply to a broader class of preference models.

**Definition 3** (BTR model). *The comparison function $\mathsf{C}$ is specified as*

$$\mathsf{C}(\tau^+, \tau^-) = \frac{\exp\left(\beta R^\star(\tau^+)\right)}{\exp\left(\beta R^\star(\tau^+)\right) + \exp\left(\beta R^\star(\tau^-)\right)},$$

*where $R^\star$ is the ground-truth reward function, $\beta > 0$ is a parameter.*

Under BTR model, the preference feedback in fact contains information of the outcome rewards. Hence, in this sense, preference-based RL can be regarded as an extension of outcome-based RL with weaker feedback.

**Algorithm for preference-based RL.** To extend Algorithm 1, we need to modify the reward model loss $\mathcal{L}_{\mathcal{D}}^{\mathsf{RM}}$ (defined in (3)) to incorporate preference feedback. For any dataset $\mathcal{D} = \{(\tau^+, \tau^-, y)\}$ consisting of (trajectories, preference) pair, we introduce the following preference-based reward model loss $\mathcal{L}_{\mathcal{D}}^{\mathsf{PbRM}}$:

$$\mathcal{L}_{\mathcal{D}}^{\mathsf{PbRM}}(R) := \sum_{(\tau^+, \tau^-, y) \in \mathcal{D}} L\left(R(\tau^+) - R(\tau^-), y\right), \tag{11}$$

where $L(w, y) := -\beta w y + \log(1 + e^{\beta w})$ is the logistic loss. It is well-known that under BTR model (Definition 3), the ground-truth reward $R^\star$ is the population minimizer of $\mathcal{L}_{\mathcal{D}}^{\mathsf{PbRM}}$, and any approximate minimizer of $\mathcal{L}_{\mathcal{D}}^{\mathsf{PbRM}}$ can serve as a proxy for $R^\star$. Therefore, with the loss function $\mathcal{L}_{\mathcal{D}}^{\mathsf{PbRM}}$, we propose the following algorithm (Algorithm 3, detailed description in Appendix E), which generalizes Algorithm 1 to handle preference feedback: For each iteration $t = 1, 2, \cdots, T$, the algorithm performs the following two steps.

1. (Optimism) Compute optimistic estimates of $(Q^\star, R^\star)$ through solving the following joint maximization problem with the dataset $\mathcal{D}$ consisting of all previously observed (trajectories, feedback) pairs:

$$(f^{(t)}, R^{(t)}) = \max_{f \in \mathcal{F}, R \in \mathcal{R}} \lambda \left[ f_1(s_1) - \widehat{V}_{\mathcal{D}, R}^{\mathrm{ref}} \right] - \mathcal{L}_{\mathcal{D}}^{\mathsf{BE}}(f; R) - \mathcal{L}_{\mathcal{D}}^{\mathsf{PbRM}}(R), \tag{12}$$

where the Bellman loss $\mathcal{L}_{\mathcal{D}}^{\mathsf{BE}}$ is defined in (5), $\widehat{V}_{\mathcal{D}, R}^{\mathrm{ref}}$ is the estimated value function of $\pi_{\mathrm{ref}}$ defined as

$$\widehat{V}_{\mathcal{D}, R}^{\mathrm{ref}} := \frac{1}{|\mathcal{D}|} \sum_{(\tau^+, \tau^-, y) \in \mathcal{D}} R(\tau^-). \tag{13}$$

The term $f_1(s_1) - \widehat{V}_{\mathcal{D}, R}^{\mathrm{ref}}$ can be regarded as an estimate of the advantage of $\pi_f$ over $\pi_{\mathrm{ref}}$ under $(f, R)$. It is introduced to avoid over-estimating the optimal value, as the preference feedback only provide information between the *difference* between two trajectories.

2. (Data collection) The algorithm selects the greedy policy $\pi^{(t)} := \pi_{f^{(t)}}$. For each $h \in [H]$, the algorithm sets $\pi^{(t,h,+)} := \pi \circ_h \pi_{\mathrm{ref}}$ and $\pi^{(t,h,-)} := \pi_{\mathrm{ref}}$, executes $(\pi^{(t,h,+)}, \pi^{(t,h,-)})$ to collects trajectories $(\tau^{(t,h,+)}, \tau^{(t,h,-)})$ and the preference feedback $y^{(t,h)}$.

We provide the following sample complexity guarantee of the algorithm above.

**Theorem 3.** *Let $\delta \in (0, 1)$. Suppose that Assumption 1 and Assumption 2 hold, and the parameters of Algorithm 3 are chosen as*

$$\lambda = c_0 \max \left\{ \frac{H \log(N_{TH^2}/\delta)}{\varepsilon}, TH\varepsilon_{\mathrm{app}} \right\}, \qquad T \geq \widetilde{O}\left( \frac{C_{\mathrm{cov}} H^2}{\varepsilon^2} \cdot \log(N_{TH^2}/\delta) \right), \qquad (14)$$

*where $c_0 > 0$ is an absolute constant, and $\widetilde{O}(\cdot)$ omits poly-logarithmic factors and constant depending on $\beta$. Then, with probability at least $1 - \delta$, the output policy $\widehat{\pi}$ of Algorithm 3 satisfies*

$$V^\star(s_1) - V^{\widehat{\pi}}(s_1) \leq \varepsilon + \widetilde{O}(C_{\mathrm{cov}} H^2 \varepsilon_{\mathrm{app}}).$$

The proof of Theorem 3 is deferred to Section E.1.

## 5 Lower Bounds

As shown by Theorem 1, with bounded coverability of the MDP and appropriate assumptions on the function classes, finding a near-optimal policy within a polynomial number of episodes with outcome rewards is possible. In this setting, our sample complexity bounds match the sample complexity of Algorithm GOLF (Jin et al., 2021a; Xie et al., 2022), up to a factor of $\widetilde{O}(H)$. This indicates that, under bounded coverability, learning with outcome-based rewards is *almost* statistically equivalent to learning with process rewards.

Additionally, in the setting of offline reinforcement learning with bounded *uniform concentrability*, the results of Jia et al. (2025) indicate that there is also a statistical equivalence between learning with outcome rewards and learning with process rewards. Therefore, it is natural to ask the following question:

> *Is learning with outcome rewards always statistically equivalent to*
> *learning with process rewards in RL?*

However, it turns out that the answer is *negative* if the statistical complexity is measured with respect to the *structure* of the value (reward) function classes.

More specifically, we construct a class of MDPs with horizon $H = 2$, *known* transition $\mathbb{T}$, and $d$-dimensional *generalized linear* reward models (Appendix F.2). With process reward feedback, such a problem is known to be *easy* as it admits low (Bellman) eluder dimension (Russo and Van Roy, 2013; Jin et al., 2021a, etc.), and existing algorithms can learn an $\varepsilon$-optimal policy using $\widetilde{O}(d^2/\varepsilon^2)$ episodes with process rewards. However, given only access to outcome rewards, we show that this problem is *as hard as* learning ReLU linear bandits (Dong et al., 2021; Li et al., 2022), and hence it requires at least $e^{\Omega(d)}$ episodes to learn. Hence, in this setting, there is an exponential separation between learning with process rewards and learning with outcome-based rewards.

**Theorem 4.** *For any positive integer $d \geq 1$, there exists a class $\mathcal{M}$ of two-layer MDPs with a fixed transition kernel $\mathbb{T}$ and initial state $s_1$, such that the following holds:*

(a) *There exists an algorithm that, for any MDP $M^\star \in \mathcal{M}$ and any $\varepsilon \in (0, 1)$, given access to process reward feedback, returns an $\varepsilon$-optimal policy with high probability using $\widetilde{O}(d^2/\varepsilon^2)$ episodes.*

(b) *Suppose that there exists an algorithm that, for any MDP $M^\star \in \mathcal{M}$, given only access to outcome reward, returns a $0.1$-optimal policy with probability at least $\frac{3}{4}$ using $T$ episodes. Then it must hold that $T = \Omega(e^{c_1 d})$, where $c_1$ is an absolute constant.* [6]

This exponential separation demonstrates that the delicate analysis based on well-behaved Bellman errors (Jiang et al., 2017; Jin et al., 2021a; Du et al., 2021, etc.) crucially relies on the process reward feedback, and the resulting guarantees might not be preserved in the setting where only outcome reward feedback is available.

---

[6] We note that under this construction, it is straightforward to construct function classes $\mathcal{F}$ and $\mathcal{R}$ such that both Assumption 1 and Assumption 2 hold, but the coverability scales as $C_{\mathrm{cov}} = e^{\Omega(d)}$.

## 6 Conclusion

In this work, we develop a model-free, sample-efficient algorithm for outcome-based reinforcement learning that relies solely on trajectory-level rewards and achieves theoretical guarantees bounded by coverability under function approximation. From the lower bound side, we show that joint optimization of reward and value functions is essential, and establish a fundamental exponential gap between outcome-based and per-step feedback. For deterministic MDPs, we propose a simpler, more efficient variant, and extend our approach to preference-based feedback, demonstrating that it preserves the same statistical efficiency. In the current work, we only studied the case where the outcome-based reward is the sum of all intermediate rewards. We leave the development of efficient algorithms for other types of outcome-based rewards to future work.

## Acknowledgements

We acknowledge support of the Simons Foundation and the NSF through awards DMS-2031883 and PHY-2019786, ARO through award W911NF-21-1-0328, and the DARPA AIQ award.

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

## A  More Related Works

We review more related works in this section.

The *coverability coefficient* has recently gained attention in the theory of online reinforcement learning (Xie et al., 2022; Liu et al., 2023a; Amortila et al., 2024a,b). This condition is in the same spirit as the widely used concentrability coefficient (Munos, 2003; Antos et al., 2008; Farahmand et al., 2010; Chen and Jiang, 2019; Jin et al., 2021b; Xie and Jiang, 2021; Xie et al., 2021; Bhardwaj et al., 2023), a concept frequently employed in the theory of offline (or batch) reinforcement learning. A well-known duality suggests that the coverability coefficient can be interpreted as the optimal concentrability coefficient attainable by any offline data distribution. For further discussion, see Xie et al. (2022).

A related body of theoretical work explores *reinforcement learning with trajectory feedback* (Neu and Bartók, 2013; Efroni et al., 2021; Chatterji et al., 2021; Cassel et al., 2024; Lancewicki and Mansour, 2025), where the learner receives only episode-level feedback at the end of each trajectory. This category also encompasses preference-based reinforcement learning (Pacchiano et al., 2021; Chen et al., 2022b; Zhu et al., 2023; Wu and Sun, 2023; Zhan et al., 2023), which relies on pairwise comparisons between trajectories. While most prior work focuses on tabular or linear MDP settings, we take a step further by studying learning with function approximation, and bound the complexity by the coverability coefficient.

In the context of *Online Reinforcement Learning*, numerous prior works have investigated the complexity of exploration and policy optimization, introducing various complexity measures such as Bellman rank (Jiang et al., 2017), Eluder dimension (Russo and Van Roy, 2013; Osband and Van Roy, 2014), witness rank (Sun et al., 2019), Bellman-Eluder dimension (Jin et al., 2021a), the bilinear class (Du et al., 2021), and decision-estimation coefficients (Foster et al., 2021). These complexity notions characterize properties of the function or model class but are generally not instance-dependent. In contrast, Xie et al. (2022) introduces the instance-dependent notion of *coverability coefficients* to provide complexity bounds in online reinforcement learning. For further discussion on instance-dependent complexity measures, we refer the reader to the discussions therein.

We further review some literatures on *online preference-based learning* or *online RLHF*. Xu et al. (2020); Novoseller et al. (2020); Pacchiano et al. (2021); Wu and Sun (2023); Zhan et al. (2023); Das et al. (2024) provides theoretical guarantees for tabular MDPs and linear MDPs. Ye et al. (2024) studies RLHF with general function approximation for contextual bandits, which is equivalent to the case where $H = 1$. Chen et al. (2022b); Wang et al. (2023) use the Eluder dimension type complexity measures to characterize the sample complexity of online RLHF, which sometimes can be too pessimistic. Xie et al. (2024); Cen et al. (2024); Zhang et al. (2024) proposed algorithms for online RLHF with function approximation, but their complexity depends on the trajectory coverability instead of the state coverability

## B  Technical tools

### B.1  Uniform convergence with square loss

To prove the uniform convergence results with square loss, we frequently use the following version Freedman's inequality (see e.g., Beygelzimer et al., 2011).

**Lemma 5** (Freedman's inequality). *Suppose that $Z^{(1)}, \cdots, Z^{(T)}$ is a martingale difference sequence that is adapted to the filtration $(\mathfrak{F}^{(t)})_{t=1}^{T}$, and $Z^{(t)} \leq C$ almost surely for all $t \in [T]$. Then for any $\lambda \in [0, \frac{1}{C}]$, with probability at least $1 - \delta$, for all $n \leq T$,*

$$\sum_{t=1}^{n} Z^{(t)} \leq \lambda \sum_{t=1}^{n} \mathbb{E}\left[ (Z^{(t)})^2 \big| \mathfrak{F}^{(t-1)} \right] + \frac{\log(1/\delta)}{\lambda}.$$

**Lemma 6.** *Suppose that $(x^{(1)}, y^{(1)}), \cdots, (x^{(T)}, y^{(T)})$ is a sequence of random variable in $\mathcal{X} \times [0, C]$ that is adapted to the filtration $(\mathfrak{F}^{(t)})_{t=1}^{T}$, such that there exists a function $F^{\star} : \mathcal{X} \to [0, 1]$ with $F^{\star}(x^{(t)}) = \mathbb{E}[y^{(t)} | \mathfrak{F}^{(t-1)}, x^{(t)}]$ almost surely. Then for any function $F : \mathcal{X} \to [0, C]$, it holds that with probability at least $1 - \delta$, for all $n \in [T]$,*

$$\sum_{t=1}^{n} \left(F(x^{(t)}) - y^{(t)}\right)^2 - \sum_{t=1}^{n} \left(F^{\star}(x^{(t)}) - y^{(t)}\right)^2 \geq \frac{1}{2} \sum_{t=1}^{n} \mathbb{E}\left[ (F(x^{(t)}) - F^{\star}(x^{(t)}))^2 \big| \mathfrak{F}^{(t-1)} \right] - 10C^2 \log(1/\delta).$$

*Conversely, it holds that with probability at least $1 - \delta$, for all $n \in [T]$,*

$$\sum_{t=1}^{n} (F(x^{(t)}) - y^{(t)})^2 - \sum_{t=1}^{n} (F^\star(x^{(t)}) - y^{(t)})^2 \leq 2 \sum_{t=1}^{n} \mathbb{E}\left[ (F(x^{(t)}) - F^\star(x^{(t)}))^2 \,\Big|\, \mathfrak{F}^{(t-1)} \right] + 5C^2 \log(1/\delta).$$

**Proof of Lemma 6.** Denote

$$\begin{aligned}
W^{(t)} &:= (F(x^{(t)}) - y^{(t)})^2 - (F^\star(x^{(t)}) - y^{(t)})^2 \\
&= (F(x^{(t)}) - F^\star(x^{(t)}))^2 + 2(F(x^{(t)}) - F^\star(x^{(t)}))(F^\star(x^{(t)}) - y^{(t)}).
\end{aligned}$$

Note that

$$\mathbb{E}\left[ W^{(t)} \,\big|\, \mathfrak{F}^{(t-1)} \right] = \mathbb{E}\left[ (F(x^{(t)}) - F^\star(x^{(t)}))^2 \,\Big|\, \mathfrak{F}^{(t-1)} \right],$$

and

$$Z^{(t)} := W^{(t)} - \mathbb{E}\left[ W^{(t)} \,\big|\, \mathfrak{F}^{(t-1)} \right] \leq W^{(t)} \leq C^2.$$

Therefore, using Freedman's inequality (Lemma 5), for any fixed $\lambda \in [0, \frac{1}{C^2}]$, we have with probability at least $1 - \delta$,

$$\sum_{t=1}^{n} Z^{(t)} \leq \lambda \sum_{t=1}^{n} \mathbb{E}\left[ (Z^{(t)})^2 \,\big|\, \mathfrak{F}^{(t-1)} \right] + \frac{\log(1/\delta)}{\lambda}, \qquad \forall n \in [T].$$

Note that

$$\begin{aligned}
\mathbb{E}\left[ (Z^{(t)})^2 \,\big|\, \mathfrak{F}^{(t-1)} \right] &\leq \mathbb{E}\left[ (W^{(t)})^2 \,\big|\, \mathfrak{F}^{(t-1)} \right] \\
&= \mathbb{E}\left[ (F(x^{(t)}) - F^\star(x^{(t)}))^4 + 4(F(x^{(t)}) - F^\star(x^{(t)}))^2 (F^\star(x^{(t)}) - y^{(t)})^2 \,\Big|\, \mathfrak{F}^{(t-1)} \right] \\
&\leq 5C^2 \mathbb{E}\left[ (F(x^{(t)}) - F^\star(x^{(t)}))^2 \,\Big|\, \mathfrak{F}^{(t-1)} \right].
\end{aligned}$$

Therefore, by setting $\lambda = \frac{1}{5C^2}$, we get the desired upper bound.

Similarly, for the lower bound, we can apply Freedman's inequality with $(-Z^{(t)})$ to show that for $\lambda = \frac{1}{10C^2}$, with probability at least $1 - \delta$,

$$\begin{aligned}
-\sum_{t=1}^{n} Z^{(t)} &\leq \lambda \sum_{t=1}^{n} \mathbb{E}\left[ (Z^{(t)})^2 \,\big|\, \mathfrak{F}^{(t-1)} \right] + \frac{\log(1/\delta)}{\lambda} \\
&\leq \frac{1}{2} \sum_{t=1}^{n} \mathbb{E}\left[ (F(x^{(t)}) - F^\star(x^{(t)}))^2 \,\Big|\, \mathfrak{F}^{(t-1)} \right] + 10C^2 \log(1/\delta), \qquad \forall n \in [T].
\end{aligned}$$

$\square$

**Proposition 7.** *Fix a parameter $\alpha \geq 0$. Under the assumption of Lemma 6, suppose that $\mathcal{H} \subseteq (\mathcal{X} \to [0, C])$ is a fixed function class, and $F^\sharp \in \mathcal{H}$ satisfies $\left\| F^\star - F^\sharp \right\|_\infty \leq \varepsilon_{\mathrm{app}}$. Define*

$$\mathcal{L}_n(F) := \sum_{t=1}^{n} (F(x^{(t)}) - y^{(t)})^2, \qquad \mathcal{E}_n(F) := \sum_{t=1}^{n} \mathbb{E}\left[ (F(x^{(t)}) - F^\star(x^{(t)}))^2 \,\Big|\, \mathfrak{F}^{(t-1)} \right].$$

*Let $\kappa := 15C^2 \log(2N(\mathcal{H}, \alpha)/\delta) + 3Cn\alpha + 4n\varepsilon_{\mathrm{app}}^2$. Then the following holds simultaneously with probability at least $1 - \delta$:*

*(1) For each $n \in [T]$,*

$$\mathcal{L}_n(F^\sharp) - \inf_{F' \in \mathcal{H}} \mathcal{L}_n(F') \leq \kappa.$$

*(2) For each $n \in [T]$, for all $F \in \mathcal{H}$,*

$$\frac{1}{2} \mathcal{E}_n(F) \leq \mathcal{L}_n(F) - \inf_{F' \in \mathcal{H}} \mathcal{L}_n(F') + \kappa.$$

**Proof of Proposition 7.**  Denote $N := N(\mathcal{H}, \alpha)$. Let $\mathcal{H}_\alpha$ be a minimal $\alpha$-covering of $\mathcal{H}$. Then applying Lemma 6 and the union bound, we have with probability at least $1 - \delta$, the following holds simultaneously for $n \in [T]$:

(1) For all $F' \in \mathcal{H}_\alpha$, it holds that

$$\frac{1}{2}\mathcal{E}_n(F') \leq \mathcal{L}_n(F') - \mathcal{L}_n(F^\star) + 10C^2 \log(2N/\delta).$$

(2) It holds that

$$\mathcal{L}_n(F^\sharp) - \mathcal{L}_n(F^\star) \leq 2\mathcal{E}_n(F^\sharp) + 5C^2 \log(2/\delta).$$

In the following, we condition on the above success event.

By definition, $\mathcal{E}_n(F^\sharp) \leq n\varepsilon_{\mathrm{app}}^2$, and hence

$$\mathcal{L}_n(F^\star) \geq \mathcal{L}_n(F^\sharp) - 4n\varepsilon_{\mathrm{app}}^2 - 5C^2 \log(2/\delta). \tag{15}$$

Furthermore, for any $F \in \mathcal{H}$, there exists $F' \in \mathcal{H}_\alpha$ with $\|F - F'\|_\infty \leq \alpha$, which implies

$$|\mathcal{L}_n(F) - \mathcal{L}_n(F')| \leq 2Cn\alpha, \qquad |\mathcal{E}_n(F) - \mathcal{E}_n(F')| \leq 2Cn\alpha.$$

Therefore, under the success event, we have

$$\frac{1}{2}\mathcal{E}_n(F) \leq \mathcal{L}_n(F) - \mathcal{L}_n(F^\star) + 10C^2 \log(2N/\delta) + 3Cn\alpha.$$

holds for arbitrary $F \in \mathcal{F}$. Hence, by (15), we have

$$\frac{1}{2}\mathcal{E}_n(F) \leq \mathcal{L}_n(F) - \mathcal{L}_n(F^\sharp) + 15C^2 \log(2N/\delta) + 3Cn\alpha + 4n\varepsilon_{\mathrm{app}}^2.$$

Noting that $\mathcal{L}_n(F^\sharp) \geq \inf_{F' \in \mathcal{H}} \mathcal{L}_n(F')$ completes the proof of (2). Furthermore, using $\mathcal{E}_n(F) \geq 0$, we also have

$$\mathcal{L}_n(F^\sharp) \leq \inf_{F' \in \mathcal{H}} \mathcal{L}_n(F') + 15C^2 \log(2N/\delta) + 3Cn\alpha + 4n\varepsilon_{\mathrm{app}}^2.$$

This completes the proof of (1). $\qquad\qquad\qquad\qquad\qquad\qquad\qquad\qquad\qquad\qquad\square$

### B.2  Uniform convergence with log-loss

We prove the following result, which is a direct extension of the standard MLE guarantee (Zhang, 2002).

**Proposition 8.** *Suppose that $\{P_\theta(y|x)\}_{\theta \in \Theta} \subseteq (\mathcal{X} \to \Delta(\mathcal{Y}))$ is a class of condition densities parametrized by an abstract parameter class $\Theta$. Without loss of generality, we assume $\mathcal{Y}$ is discrete.*

*A $\alpha$-covering of $\Theta$ is a subset $\Theta' \subseteq \Theta$ such that for any $\theta \in \Theta$, there exists $\theta' \in \Theta'$ such that $|\log P_\theta(y|x) - \log P_{\theta'}(y|x)| \leq \alpha$ for all $x \in \mathcal{X}, y \in \mathcal{Y}$. We define the covering number of $\Theta$ under log-loss as*

$$N_{\log}(\Theta, \alpha) := \min\{|\Theta'| : \Theta' \text{ is a } \alpha\text{-covering of } \Theta\}.$$

*Suppose that $(x^{(1)}, y^{(1)}), \cdots, (x^{(T)}, y^{(T)})$ is a sequence of random variables adapted to the filtration $(\mathfrak{F}^{(t)})_{t=1}^T$, such that there exists $\theta^\star \in \Theta$ so that $\mathbb{P}(y^{(t)} = \cdot|x^{(t)}, \mathfrak{F}^{(t-1)}) = P_{\theta^\star}(y^{(t)} = \cdot|x^{(t)})$ almost surely for $t \in [T]$. Then it holds that for all $n \in [T]$, for all $\theta \in \Theta$,*

$$\sum_{t=1}^n \mathbb{E}\left[D_{\mathrm{H}}^2(P_\theta(\cdot|x^{(t)}), P_{\theta^\star}(\cdot|x^{(t)})) \,\middle|\, \mathfrak{F}^{(t-1)}\right] \leq -\frac{1}{2}\sum_{t=1}^n \left[\log P_\theta(y^{(t)}|x^{(t)}) - \log P_{\theta^\star}(y^{(t)}|x^{(t)})\right]$$
$$+ \log N_{\log}(\Theta, \alpha) + 2n\alpha.$$

**Proof of Proposition 8.**  Let $\Theta' \subseteq \Theta$ be a minimal $\alpha$-covering, and let $N := |\Theta'| = N_{\log}(\Theta, \alpha)$. For each $\theta \in \Theta$, we consider

$$L^{(t)}(\theta) := \log P_\theta(y^{(t)}|x^{(t)}) - \log P_{\theta^\star}(y^{(t)}|x^{(t)}).$$

Then it holds that

$$\mathbb{E}\left[\exp\left(\frac{1}{2}L^{(t)}(\theta)\right)\Big|\, x^{(t)}, \mathfrak{F}^{(t-1)}\right] = \mathbb{E}_{y \sim P_{\theta^\star}(\cdot|x^{(t)})}\sqrt{\frac{P_\theta(y|x^{(t)})}{P_{\theta^\star}(y|x^{(t)})}}$$

$$= \sum_{y \in \mathcal{Y}}\sqrt{P_\theta(y|x^{(t)})P_{\theta^\star}(y|x^{(t)})}$$

$$= 1 - D_{\mathrm{H}}^2(P_\theta(\cdot|x^{(t)}), P_{\theta^\star}(\cdot|x^{(t)})).$$

Therefore, applying Lemma 9 and using union bound over $\theta \in \Theta'$, we have the following bound: with probability at least $1 - \delta$, for any $\theta' \in \Theta'$, $n \in [T]$,

$$\sum_{t=1}^{n} -\log\left[\exp\left(\frac{1}{2}L^{(t)}(\theta')\right)\Big|\, \mathcal{F}^{(t-1)}\right] \leq -\frac{1}{2}\sum_{t=1}^{n}L^{(t)}(\theta') + \log(N/\delta).$$

In the following, we condition on the above event. Fix any $\theta \in \Theta$. Then, there exists $\theta' \in \Theta'$ such that $|\log P_\theta(y|x) - \log P_{\theta'}(y|x)| \leq \alpha$ for all $x \in \mathcal{X}, y \in \mathcal{Y}$, and hence $|L^{(t)}(\theta) - L^{(t)}(\theta')| \leq \alpha$ almost surely. Therefore, combining the results above and using $\log w \leq w - 1$ for $w > 0$, we have

$$\sum_{t=1}^{n}\mathbb{E}\left[D_{\mathrm{H}}^2(P_\theta(\cdot|x^{(t)}), P_{\theta^\star}(\cdot|x^{(t)}))\big|\,\mathfrak{F}^{(t-1)}\right] \leq \sum_{t=1}^{n} -\log\left[\exp\left(\frac{1}{2}L^{(t)}(\theta)\right)\Big|\, \mathcal{F}^{(t-1)}\right]$$

$$\leq n\alpha + \sum_{t=1}^{n} -\log\left[\exp\left(\frac{1}{2}L^{(t)}(\theta')\right)\Big|\, \mathcal{F}^{(t-1)}\right]$$

$$\leq n\alpha - \frac{1}{2}\sum_{t=1}^{n}L^{(t)}(\theta') + \log(N/\delta)$$

$$\leq 2n\alpha - \frac{1}{2}\sum_{t=1}^{n}L^{(t)}(\theta) + \log(N/\delta).$$

By the arbitrariness of $\theta \in \Theta$, the proof is completed. $\qquad\square$

**Lemma 9** (Foster et al. (2021, Lemma A.4)). *For any sequence of real-valued random variables* $X^{(1)}, \cdots, X^{(T)}$ *adapted to a filtration* $(\mathfrak{F}^{(t)})_{t=1}^{T}$, *it holds that with probability at least* $1 - \delta$, *for all* $n \in [T]$,

$$\sum_{t=1}^{n} -\log\left[\exp(-X^{(t)})|\,\mathcal{F}^{(t-1)}\right] \leq \sum_{t=1}^{n}X^{(t)} + \log\left(1/\delta\right).$$

## C    Missing Proofs in Section 3.1

### C.1    Proof of Theorem 1

We first present a more detailed statement of the upper bound of Theorem 1, as follows.

**Theorem 10.** *Let* $\delta \in (0,1)$, $\rho \in [0,1)$, *and we denote* $C_{\mathrm{cov}} = C_{\mathrm{cov}}(\Pi_{\mathcal{F}})$, *where* $\Pi_{\mathcal{F}} = \{\pi_f : f \in \mathcal{F}\}$ *is the policy class induced by* $\mathcal{F}$. *Suppose that Assumption 1 and Assumption 2 hold. Then with probability at least* $1 - \delta$, *the output policy* $\widehat{\pi}$ *of Algorithm 1 satisfies*

$$V^\star(s_1) - V^{\widehat{\pi}}(s_1) = \frac{1}{T}\sum_{t=1}^{T}\left(V^\star(s_1) - V^{\pi^{(t)}}(s_1)\right)$$

$$\leq O(H) \cdot \left[\frac{\log(N(\rho)/\delta) + TH^2(\rho + \varepsilon_{\mathrm{app}}^2)}{\lambda} + \frac{\lambda C_{\mathrm{cov}}\log(T)}{T}\right].$$

*Therefore, for any* $\varepsilon \in (0,1)$, *with the optimally-tuned parameter* $\lambda$, *it holds that* $V^\star(s_1) - V^{\widehat{\pi}}(s_1) \leq \varepsilon + \widetilde{O}\big(\sqrt{C_{\mathrm{cov}}}H^2\varepsilon_{\mathrm{app}}\big)$, *as long as*

$$T \geq \widetilde{O}\left(\frac{C_{\mathrm{cov}}H^2}{\varepsilon^2} \cdot \log N(\varepsilon^2/(C_{\mathrm{cov}}H^4))\right).$$

Recall that we let $Q^\sharp \in \mathcal{Q}, R^\sharp \in \mathcal{R}$ be such that

$$\max_{h \in [H]} \left\| Q_h^\sharp - Q_h^\star \right\|_\infty \leq \varepsilon_{\text{app}}, \qquad \max_{h \in [H]} \left\| R_h^\sharp - R_h^\star \right\|_\infty \leq \varepsilon_{\text{app}}.$$

For each $t \in [T]$, we write $\mathcal{D}^{(t)}$ to be the dataset maintained by Algorithm 1 at the end of the $t$th iteration, i.e.,

$$\mathcal{D}^{(t)} = \{(\tau^{(k,h)}, r^{(k,h)})\}_{k \leq t, h \in [H]}.$$

We summarize the uniform concentration results for the loss $\mathcal{L}_{\mathcal{D}^{(t)}}^{\mathsf{BE}}$ and $\mathcal{L}_{\mathcal{D}^{(t)}}^{\mathsf{RM}}$ as follows. We note that these concentration bounds are fairly standard (see e.g. Jin et al. (2021a)), and for completeness, we present the proof in Appendix C.3.

**Proposition 11.** *Let $\delta \in (0,1), \rho \geq 0$. Suppose that Assumption 1 and Assumption 2 holds. Then with probability at least $1 - \delta$, for all $t \in [T], f \in \mathcal{F}, R \in \mathcal{R}$, it holds that*

$$\frac{1}{2} \sum_{k \leq t} \sum_{h=1}^H \mathbb{E}^{\pi^{(k)} \circ_h \pi_{\text{ref}}} (R(\tau) - R^\star(\tau))^2 \leq \mathcal{L}_{\mathcal{D}^{(t)}}^{\mathsf{RM}}(R) - \mathcal{L}_{\mathcal{D}^{(t)}}^{\mathsf{RM}}(R^\sharp) + H\kappa,$$

$$\frac{1}{2} \sum_{k \leq t} \sum_{h=1}^H \mathbb{E}^{\pi^{(k)}} (f_h(s_h, a_h) - [\mathcal{T}_{R,h} f_{h+1}](s_h, a_h))^2 \leq \mathcal{L}_{\mathcal{D}^{(t)}}^{\mathsf{BE}}(f; R) - \mathcal{L}_{\mathcal{D}^{(t)}}^{\mathsf{BE}}(Q^\sharp; R^\sharp) + H\kappa,$$

*where*

$$\kappa = C\big(\log N(\rho) + \log(H/\delta) + TH^2(\varepsilon_{\text{app}}^2 + \rho)\big),$$

*and $C > 0$ is an absolute constant.*

**Performance difference decomposition.** Denote $V^\sharp(s_1) := \max_{a \in \mathcal{A}} Q^\sharp(s_1, a)$. Then it is clear that $\left| V^\sharp(s_1) - V^\star(s_1) \right| \leq \varepsilon_{\text{app}}$. Therefore, for any $t \in [T]$, by optimism (the definition of $(f^{(t)}, R^{(t)})$), it holds that

$$\begin{aligned}
V^\star(s_1) - \varepsilon_{\text{app}} &\leq V^\sharp(s_1) \\
&= V^\sharp(s_1) - \frac{\mathcal{L}_{\mathcal{D}^{(t-1)}}^{\mathsf{BE}}(Q^\sharp, R^\sharp) + \mathcal{L}_{\mathcal{D}^{(t-1)}}^{\mathsf{RM}}(R^\sharp)}{\lambda} + \frac{\mathcal{L}_{\mathcal{D}^{(t-1)}}^{\mathsf{BE}}(Q^\sharp, R^\sharp) + \mathcal{L}_{\mathcal{D}^{(t-1)}}^{\mathsf{RM}}(R^\sharp)}{\lambda} \\
&\leq f_1^{(t)}(s_1, \pi^{(t)}) - \frac{\mathcal{L}_{\mathcal{D}^{(t-1)}}^{\mathsf{BE}}(f^{(t)}, R^{(t)}) + \mathcal{L}_{\mathcal{D}^{(t-1)}}^{\mathsf{RM}}(R^{(t)})}{\lambda} + \frac{\mathcal{L}_{\mathcal{D}^{(t-1)}}^{\mathsf{BE}}(Q^\sharp, R^\sharp) + \mathcal{L}_{\mathcal{D}^{(t-1)}}^{\mathsf{RM}}(R^\sharp)}{\lambda}.
\end{aligned}$$

Furthermore, by the standard performance difference lemma (Kakade and Langford, 2002), it holds that

$$f_1^{(t)}(s_1, \pi^{(t)}) - V^{\pi^{(t)}}(s_1) = \sum_{h=1}^H \mathbb{E}^{\pi^{(t)}} \big[ f_h^{(t)}(s_h, a_h) - [\mathcal{T}_h^\star f_{h+1}^{(t)}](s_h, a_h) \big]. \tag{16}$$

Based on (16), the existing approaches (with per-step reward feedback) bound the expectation of the Bellman error $e_h^{(t)}(s_h, a_h) := f_h^{(t)}(s_h, a_h) - [\mathcal{T}_h^\star f_{h+1}^{(t)}](s_h, a_h)$ through various arguments (e.g., eluder argument (Jin et al., 2021a) and coverability argument (Xie et al., 2022)). However, in outcome reward model, it is possible that $\mathbb{E}^{\pi^{(t)}} \big[ f_h^{(t)}(s_h, a_h) - [\mathcal{T}_h^\star f_{h+1}^{(t)}](s_h, a_h) \big]$ is large even when the sub-optimality of $\pi^{(t)}$ is small, because the outcome reward is invariant under shifting of the ground-truth reward function $R^\star$.

Therefore, we consider the following refined decomposition:

$$\begin{aligned}
f_1^{(t)}(s_1) - V^{\pi^{(t)}}(s_1) = &\sum_{h=1}^H \mathbb{E}^{\pi^{(t)}} \big[ f_h^{(t)}(s_h, a_h) - [\mathcal{T}_{R^{(t)}} f_{h+1}^{(t)}](s_h, a_h) \big] \\
&+ \mathbb{E}^{\pi^{(t)}} \bigg[ \sum_{h=1}^H R_h^{(t)}(s_h, a_h) - \sum_{h=1}^H R_h^\star(s_h, a_h) \bigg].
\end{aligned} \tag{17}$$

**Coverability argument.** Following the coverability argument of Xie et al. (2022), we have the following upper bound on the expected Bellman errors.

**Proposition 12.** *Denote* $e_h^{(t)} := f_h^{(t)} - \mathcal{T}_h^\star f_{h+1}$. *Then, for each* $h \in [H]$*, it holds that*

$$\sum_{t=1}^{T} \mathbb{E}^{\pi^{(t)}} \left| e_h^{(t)}(s_h, a_h) \right| \leq \sqrt{2C_{\text{cov}} \log \left( 1 + \frac{C_{\text{cov}} T}{\kappa} \right) \cdot \left[ 2T\kappa + \sum_{1 \leq k < t \leq T} \mathbb{E}^{\pi^{(k)}} e_h^{(t)}(s_h, a_h)^2 \right]}.$$

Following the ideas of Jia et al. (2025), we prove the following upper bound on the reward errors.

**Proposition 13.** *It holds that*

$$\sum_{t=1}^{T} \mathbb{E}^{\pi^{(t)}} \left| R^{(t)}(\tau) - R^\star(\tau) \right|$$

$$\leq \sqrt{8HC_{\text{cov}} \log \left( 1 + \frac{C_{\text{cov}} T}{\kappa} \right) \cdot \left[ HT\kappa + \sum_{1 \leq k < t \leq T} \sum_{h=1}^{H} \mathbb{E}^{\pi^{(k)} \circ_h \pi_{\text{ref}}} (R^{(t)}(\tau) - R^\star(\tau))^2 \right]}.$$

Based on the results above, we finalize the proof of Theorem 1.

**Proof of Theorem 1.** By optimism and the decomposition (17), it holds that for $t \in [T]$,

$$V^\star(s_1) - V^{\pi^{(t)}}(s_1) - \varepsilon_{\text{app}} \leq \sum_{h=1}^{H} \mathbb{E}^{\pi^{(t)}} \left[ e_h^{(t)}(s_h, a_h) \right] - \frac{\mathcal{L}_{\mathcal{D}^{(t-1)}}^{\text{BE}}(f^{(t)}, R^{(t)}) - \mathcal{L}_{\mathcal{D}^{(t-1)}}^{\text{BE}}(Q^\sharp, R^\sharp)}{\lambda}$$

$$+ \mathbb{E}^{\pi^{(t)}} [R^{(t)}(\tau) - R^\star(\tau)] - \frac{\mathcal{L}_{\mathcal{D}^{(t-1)}}^{\text{RM}}(R^{(t)}) - \mathcal{L}_{\mathcal{D}^{(t-1)}}^{\text{RM}}(R^\sharp)}{\lambda}.$$

Then, under the success event of Proposition 11, we have

$$V^\star(s_1) - V^{\pi^{(t)}}(s_1) - \varepsilon_{\text{app}} - \frac{2H\kappa}{\lambda}$$

$$\leq \sum_{h=1}^{H} \left( \mathbb{E}^{\pi^{(t)}} \left[ e_h^{(t)}(s_h, a_h) \right] - \frac{1}{2\lambda} \sum_{k<t} \mathbb{E}^{\pi^{(k)}} e_h^{(t)}(s_h, a_h)^2 \right) \tag{18}$$

$$+ \mathbb{E}^{\pi^{(t)}} [R^{(t)}(\tau) - R^\star(\tau)] - \frac{1}{2\lambda} \sum_{k<t} \sum_{h=1}^{H} \mathbb{E}^{\pi^{(k)} \circ_h \pi_{\text{ref}}} (R^{(t)}(\tau) - R^\star(\tau))^2.$$

Applying Proposition 12 and Cauchy inequality gives for all $h \in [H]$,

$$\sum_{t=1}^{T} \mathbb{E}^{\pi^{(t)}} \left| e_h^{(t)}(s_h, a_h) \right| \leq \lambda C_{\text{cov}} \log \left( 1 + \frac{C_{\text{cov}} T}{\kappa} \right) + \frac{1}{2\lambda} \left[ 2T\kappa + \sum_{1 \leq k < t \leq T} \mathbb{E}^{\pi^{(k)}} e_h^{(t)}(s_h, a_h)^2 \right],$$

and similarly, applying Proposition 13 and Cauchy inequality gives

$$\sum_{t=1}^{T} \mathbb{E}^{\pi^{(t)}} \left| R^{(t)}(\tau) - R^\star(\tau) \right|$$

$$\leq 4\lambda H C_{\text{cov}} \log \left( 1 + \frac{C_{\text{cov}} T}{\kappa} \right) + \frac{1}{2\lambda} \left[ HT\kappa + \sum_{1 \leq k < t \leq T} \sum_{h=1}^{H} \mathbb{E}^{\pi^{(k)} \circ_h \pi_{\text{ref}}} (R^{(t)}(\tau) - R^\star(\tau))^2 \right].$$

Therefore, we take summation of (18) over $t \in [T]$, and combining the inequalities above gives

$$\sum_{t=1}^{T} \left( V^\star(s_1) - V^{\pi^{(t)}}(s_1) \right) \leq T\varepsilon_{\text{app}} + \frac{4TH\kappa}{\lambda} + 5\lambda H C_{\text{cov}} \log \left( 1 + \frac{C_{\text{cov}} T}{\kappa} \right).$$

This is the desired upper bound. □

## C.2 Proof of Proposition 12 and Proposition 13

The following proposition is an generalized version of the results in Xie et al. (2022, Appendix D). For proof, see e.g. Chen et al. (2024).

**Proposition 14** (Xie et al. (2022)). *Let $C \geq 1$ be a parameter. Suppose that $p^{(1)}, \cdots, p^{(T)}$ is a sequence of distributions over $\mathcal{X}$, and there exists $\mu \in \Delta(\mathcal{X})$ such that $p^{(t)}(x)/\mu(x) \leq C$ for all $x \in \mathcal{X}, t \in [T]$. Then for any sequence $\psi^{(1)}, \cdots, \psi^{(T)}$ of functions $\mathcal{X} \to [0,1]$ and constant $B \geq 1$, it holds that*

$$\sum_{t=1}^{T} \mathbb{E}_{x \sim p^{(t)}} \psi^{(t)}(x) \leq \sqrt{2C \log\left(1 + \frac{CT}{B}\right)\left[2TB + \sum_{t=1}^{T} \sum_{k<t} \mathbb{E}_{x \sim p^{(k)}} \psi^{(t)}(x)^2\right]}.$$

As a warm-up, we prove Proposition 12 by directly invoking Proposition 14.

**Proof of Proposition 12.** Fix a $h \in [H]$. To apply Proposition 14, we consider $\mathcal{X} = \mathcal{S} \times \mathcal{A}$, and define

$$p^{(t)} := \mathbb{P}^{\pi^{(t)}}((s_h, a_h) = \cdot) \in \Delta(\mathcal{S} \times \mathcal{A}), \qquad \psi^{(t)} := \left|e_h^{(t)}\right| \in (\mathcal{S} \times \mathcal{A} \to [0,1]).$$

By the definition of coverability (Definition 1), for $C = C_{\mathrm{cov}}(\Pi)$, there exists $\mu \in \Delta(\mathcal{S} \times \mathcal{A})$ such that $p^{(t)}(s,a)/\mu(s,a) \leq C$ for all $(s,a) \in \mathcal{S} \times \mathcal{A}$. Therefore, applying Proposition 14 with $B = \kappa \geq 1$ gives the desired upper bound. $\square$

Next, we proceed to prove Proposition 13. Our key proof technique is summarized in the following proposition, which is inspired by the (rather sophisticated) analysis of Jia et al. (2025).

**Proposition 15.** *Recall that for any $D = (D_h : \mathcal{S} \times \mathcal{A} \to \mathbb{R})$, we denote*

$$D(\tau) = \sum_{h=1}^{H} D_h(s_h, a_h), \qquad \forall \tau = (s_1, a_1, \cdots, s_H, a_H) \in (\mathcal{S} \times \mathcal{A})^H.$$

*Fix a Markov policy $\pi_{\mathrm{ref}}$, we denote $\overline{D}_1(s) = \overline{D}_{H+1}(s) = 0$, and*

$$\overline{D}_h(s) := \mathbb{E}^{\pi_{\mathrm{ref}}}\left[\sum_{\ell=h}^{H} D_\ell(s_\ell, a_\ell) \middle| s_h = s\right], \qquad \forall 1 < h \leq H, s \in \mathcal{S}.$$

*Then for any policy $\pi$, it holds that*

$$\sum_{h=1}^{H} \mathbb{E}^{\pi}\left(D_h(s_h, a_h) + \overline{D}_{h+1}(s_{h+1}) - \overline{D}_h(s_h)\right)^2 \leq 4 \sum_{h=1}^{H} \mathbb{E}^{\pi \circ_h \pi_{\mathrm{ref}}} D(\tau)^2.$$

The proof of Proposition 15 is deferred to the end of this subsection. With Proposition 15, we prove Proposition 13 as follows.

**Proof of Proposition 13.** To apply Proposition 15, for each $t \in [T]$, we consider

$$\Delta_h^{(t)}(s) := \mathbb{E}^{\pi_{\mathrm{ref}}}\left[\sum_{\ell=h}^{H} R_\ell^{(t)}(s_\ell, a_\ell) - \sum_{\ell=h}^{H} R_\ell^{\star}(s_\ell, a_\ell) \middle| s_h = s\right], \quad \forall h = 2, \cdots, H, s \in \mathcal{S},$$

Then, by Proposition 15, it holds that for any policy $\pi$,

$$\sum_{h=1}^{H} \mathbb{E}^{\pi}\left(R_h^{(t)}(s_h, a_h) - R_h^{\star}(s_h, a_h) + \Delta_{h+1}^{(t)}(s_{h+1}) - \Delta_h^{(t)}(s_h)\right)^2$$

$$\leq 4 \sum_{h=1}^{H} \mathbb{E}^{\pi \circ_h \pi_{\mathrm{ref}}}\left(R^{(t)}(\tau) - R^{\star}(\tau)\right)^2. \tag{19}$$

Furthermore,

$$
\mathbb{E}^{\pi}\left|R^{(t)}(\tau) - R^{\star}(\tau)\right| = \mathbb{E}^{\pi}\left|\sum_{h=1}^{H}\left[R_h^{(t)}(s_h, a_h) - R_h^{\star}(s_h, a_h) + \Delta_{h+1}^{(t)}(s_{h+1}) - \Delta_h^{(t)}(s_h)\right]\right|
$$
$$
\leq \sum_{h=1}^{H}\mathbb{E}^{\pi}\left|R_h^{(t)}(s_h, a_h) - R_h^{\star}(s_h, a_h) + \Delta_{h+1}^{(t)}(s_{h+1}) - \Delta_h^{(t)}(s_h)\right|.
\tag{20}
$$

Therefore, to apply Proposition 14, we consider the space $\mathcal{X} = \mathcal{S} \times \mathcal{A} \times \mathcal{S}$, and for each $h \in [H]$, we define

$$
p_h^{(t)} := \mathbb{P}^{\pi^{(t)}}((s_h, a_h, s_{h+1}) = \cdot) \in \Delta(\mathcal{S} \times \mathcal{A} \times \mathcal{S}),
$$
$$
\psi_h^{(t)}(s, a, s') := \left|R_h^{(t)}(s, a) - R_h^{\star}(s, a) + \Delta_{h+1}^{(t)}(s') - \Delta_h^{(t)}(s)\right|.
$$

Note that for any $h$, we have $\psi_h^{(t)} : \mathcal{S} \times \mathcal{A} \times \mathcal{S} \to [0, 1]$. Further, let $\mu_h \in \Delta(\mathcal{S} \times \mathcal{A})$ be the distribution such that $\|d_h^{\pi}/\mu_h\|_{\infty} \leq C_{\mathrm{cov}}$ for any policy $\pi$. Then we can consider the distribution $\overline{\mu}_h \in \Delta(\mathcal{S} \times \mathcal{A} \times \mathcal{S})$ given by $\overline{\mu}_h(s, a, s') = \mu_h(s, a)\mathbb{T}_h(s'|s, a)$. Then it holds that

$$
\left\|\frac{p_h^{(t)}}{\overline{\mu}_h}\right\|_{\infty} = \sup_{s,a,s'}\frac{p_h^{(t)}(s, a, s')}{\overline{\mu}_h(s, a, s')} = \sup_{s,a,s'}\frac{d_h^{\pi^{(t)}}(s, a)\mathbb{T}_h(s'|s, a)}{\mu_h(s, a)\mathbb{T}_h(s'|s, a)} \leq C_{\mathrm{cov}}.
$$

Therefore, for $h \in [H]$, applying Proposition 14 on the sequence $(p_h^{(1)}, \cdots, p_h^{(T)})$ and $(\psi_h^{(1)}, \cdots, \psi_h^{(T)})$ gives

$$
\sum_{t=1}^{T}\mathbb{E}_{x \sim p_h^{(t)}}\psi_h^{(t)}(x) \leq \sqrt{2C_{\mathrm{cov}}\log\left(1 + \frac{C_{\mathrm{cov}}T}{\kappa}\right)\left[2T\kappa + \sum_{t=1}^{T}\sum_{k<t}\mathbb{E}_{x \sim p_h^{(k)}}\psi_h^{(t)}(x)^2\right]}.
$$

To conclude, we combine the inequalities above and bound

$$
\sum_{t=1}^{T}\mathbb{E}^{\pi^{(t)}}\left|R^{(t)}(\tau) - R^{\star}(\tau)\right|
$$
$$
\leq \sum_{t=1}^{T}\sum_{h=1}^{H}\mathbb{E}^{\pi^{(t)}}\left|R_h^{(t)}(s_h, a_h) - R_h^{\star}(s_h, a_h) + \Delta_{h+1}^{(t)}(s_{h+1}) - \Delta_h^{(t)}(s_h)\right|
$$
$$
= \sum_{t=1}^{T}\sum_{h=1}^{H}\mathbb{E}_{x \sim p_h^{(t)}}\psi_h^{(t)}(x) = \sum_{h=1}^{H}\sum_{t=1}^{T}\mathbb{E}_{x \sim p_h^{(t)}}\psi_h^{(t)}(x)
$$
$$
\leq \sum_{h=1}^{H}\sqrt{2C_{\mathrm{cov}}\log\left(1 + \frac{C_{\mathrm{cov}}T}{\kappa}\right)\left[2T\kappa + \sum_{t=1}^{T}\sum_{k<t}\mathbb{E}_{x \sim p_h^{(k)}}\psi_h^{(t)}(x)^2\right]}
$$
$$
\leq \sqrt{2HC_{\mathrm{cov}}\log\left(1 + \frac{C_{\mathrm{cov}}T}{\kappa}\right)\left[2TH\kappa + \sum_{h=1}^{H}\sum_{t=1}^{T}\sum_{k<t}\mathbb{E}_{x \sim p_h^{(k)}}\psi_h^{(t)}(x)^2\right]}
$$
$$
\leq \sqrt{8HC_{\mathrm{cov}}\log\left(1 + \frac{C_{\mathrm{cov}}T}{\kappa}\right) \cdot \left[HT\kappa + \sum_{1 \leq k < t \leq T}\sum_{h=1}^{H}\mathbb{E}^{\pi^{(k)} \circ_h \pi_{\mathrm{ref}}}\left(R^{(t)}(\tau) - R^{\star}(\tau)\right)^2\right]},
$$

where the first inequality follows from (20), the second last line follows from Cauchy inequality, and last inequality follows from the definition of $(p_h^{(t)}, \psi_h^{(t)})$ and (19). $\qquad \square$

**Proof of Proposition 15.** For $\tau = (s_1, a_1, \cdots, s_H, a_H) \in (\mathcal{S} \times \mathcal{A})^H$, we denote $\tau_h = (s_1, a_1, \cdots, s_h, a_h, s_{h+1})$ to be the prefix sequence of $\tau$ for each $h \in [H]$. Then, we note that

$$
\mathbb{E}^{\pi \circ_h \pi_{\mathrm{ref}}}\left[D(\tau)|\tau_h\right] = \sum_{\ell=1}^{h}D_\ell(s_\ell, a_\ell) + \mathbb{E}^{\pi \circ_h \pi_{\mathrm{ref}}}\left[\sum_{\ell=h+1}^{H}D_\ell(s_\ell, a_\ell)\,\middle|\,\tau_h\right]
$$

$$= \sum_{\ell=1}^{h} D_\ell(s_\ell, a_\ell) + \overline{D}_{h+1}(s_{h+1}),$$

because the policy $\pi \circ_h \pi_{\mathrm{ref}}$ executes the Markov policy $\pi_{\mathrm{ref}}$ starting at the $(h+1)$-th step. Therefore, it holds that

$$\mathbb{E}_{\tau_h \sim \pi}\left(\sum_{\ell=1}^{h} D_\ell(s_\ell, a_\ell) + \overline{D}_{h+1}(s_{h+1})\right)^2 = \mathbb{E}_{\tau_h \sim \pi \circ_h \pi_{\mathrm{ref}}}(\mathbb{E}^{\pi \circ_h \pi_{\mathrm{ref}}}\left[D(\tau) \,|\, \tau_h\right])^2$$

$$\leq \mathbb{E}_{\tau \sim \pi \circ_h \pi_{\mathrm{ref}}} D(\tau)^2 = \mathbb{E}^{\pi \circ_h \pi_{\mathrm{ref}}} D(\tau)^2,$$

where the first equality follows from the fact that the policy $\pi \circ_h \pi_{\mathrm{ref}}$ executes $\pi$ for the first $h$ steps. Therefore, for $h > 1$, it holds that

$$\mathbb{E}^\pi\big(D_h(s_h, a_h) + \overline{D}_{h+1}(s_{h+1}) - \overline{D}_h(s_h)\big)^2$$

$$= \mathbb{E}_{\tau_h \sim \pi}\left(\sum_{\ell=1}^{h} D_\ell(s_\ell, a_\ell) + \overline{D}_{h+1}(s_{h+1}) - \sum_{\ell=1}^{h-1} D_\ell(s_\ell, a_\ell) - \overline{D}_h(s_h)\right)^2$$

$$\leq 2\mathbb{E}_{\tau_h \sim \pi}\left(\sum_{\ell=1}^{h} D_\ell(s_\ell, a_\ell) + \overline{D}_{h+1}(s_{h+1})\right)^2 + \mathbb{E}_{\tau_{h-1} \sim \pi}\left(\sum_{\ell=1}^{h-1} D_\ell(s_\ell, a_\ell) + \overline{D}_h(s_h)\right)^2$$

$$\leq 2\mathbb{E}^{\pi \circ_h \pi_{\mathrm{ref}}} D(\tau)^2 + \mathbb{E}^{\pi \circ_{h-1} \pi_{\mathrm{ref}}} D(\tau)^2.$$

For $h = 1$, because $\overline{D}_1(s) = 0$, we already have

$$\mathbb{E}^\pi\big(D_1(s_1, a_1) + \overline{D}_2(s_2) - \overline{D}_1(s_1)\big)^2 = \mathbb{E}^\pi\big(D_1(s_1, a_1) + \overline{D}_2(s_2)\big)^2 \leq \mathbb{E}^{\pi \circ_1 \pi_{\mathrm{ref}}} D(\tau)^2.$$

Taking summation over $h \in [H]$ completes the proof. $\qquad\square$

## C.3 Proof of Proposition 11

We prove Proposition 11 in Lemma 16 and Lemma 17 separately. Recall again that under Assumption 1, the function $Q^\sharp \in \mathcal{F}, R^\sharp \in \mathcal{R}$ satisfy

$$\max_{h \in [H]} \left\|Q_h^\sharp - Q_h^\star\right\|_\infty \leq \varepsilon_{\mathrm{app}}, \qquad \max_{h \in [H]} \left\|R_h^\sharp - R_h^\star\right\|_\infty \leq \varepsilon_{\mathrm{app}}.$$

**Lemma 16.** *Under Assumption 1, with probability at least $1 - \delta$, for any $t \in [T]$, for all $R \in \mathcal{R}$, it holds that*

$$\frac{1}{2} \sum_{k \leq t} \sum_{h=1}^{H} \mathbb{E}^{\pi^{(k)} \circ_h \pi_{\mathrm{ref}}}(R(\tau) - R^\star(\tau))^2 \leq \mathcal{L}_{\mathcal{D}^{(t)}}^{\mathsf{RM}}(R) - \mathcal{L}_{\mathcal{D}^{(t)}}^{\mathsf{RM}}(R^\sharp)$$

$$+ 15H \log N(\rho) + 15(2/\delta) + 2TH^3 \varepsilon_{\mathrm{app}}^2 + 4TH^2 \rho.$$

**Proof of Lemma 16.** To apply Proposition 7, we consider the whole history

$$\{(\tau^{(t,h)}, r^{(t,h)})\}_{t \in [T], h \in [H]}.$$

generated by executing Algorithm 1, and recall that $\mathcal{D}^{(t-1)} = \{(\tau^{(k,h)}, r^{(k,h)})\}_{k < t, h \in [H]}$ is the history up to the $t$-th iteration. Note that $(\tau^{(t,1)}, r^{(t,1)}), \cdots, (\tau^{(t,H)}, r^{(t,H)})$ are pairwise independent given $\mathcal{D}^{(t-1)}$, with

$$\tau^{(t,h)} \sim \pi^{(t)} \circ_h \pi_{\mathrm{ref}}, \qquad \mathbb{E}\left[r^{(t,h)} \,|\, \mathcal{D}^{(t-1)}, \tau^{(t,h)}\right] = R^\star(\tau^{(t,h)}).$$

Also note that $r \in [0, 1]$ almost surely, and we regard $\mathcal{R} \subseteq ((\mathcal{S} \times \mathcal{A})^H \to [0, 1])$, and $R^\sharp \in \mathcal{R}$ satisfies $\left|R^\sharp(\tau) - R^\star(\tau)\right| \leq H \varepsilon_{\mathrm{app}}$ for all $\tau \in (\mathcal{S} \times \mathcal{A})^H$.

Therefore, applying Proposition 7 on the function class $\mathcal{R}$ and the sequence $\{(\tau^{(t,h)}, r^{(t,h)})\}_{t \in [T], h \in [0,H]}$ gives that with probability at least $1 - \delta$, for all $R \in \mathcal{R}, t \in [T]$, it holds that

$$\frac{1}{2} \sum_{k=1}^{t} \sum_{h=1}^{H} \mathbb{E}^{\pi^{(k)} \circ_h \pi_{\mathrm{ref}}}(R(\tau) - R^\star(\tau))^2 = \frac{1}{2} \sum_{k=1}^{t} \sum_{h=1}^{H} \mathbb{E}\left[(R(\tau^{(k,h)}) - R^\star(\tau^{(k,h)}))^2 \,\Big|\, \mathcal{D}^{(k-1)}\right]$$

$$\leq \mathcal{L}_{\mathcal{D}^{(t)}}^{\mathsf{RM}}(R) - \mathcal{L}_{\mathcal{D}^{(t)}}^{\mathsf{RM}}(R^\sharp) + 15\log(2N(\mathcal{R}, H\rho)/\delta) + 2TH^3\varepsilon_{\mathrm{app}}^2 + 4TH^2\rho.$$

Finally, we note that $\log N(\mathcal{R}, H\rho) \leq \sum_{h=1}^{H} \log N(\mathcal{R}_h, \rho) \leq H\log N(\rho)$. This gives the desired upper bound. $\qquad\square$

Similarly, we prove Proposition 11 (2) as follows, following Jin et al. (2021a).

**Lemma 17.** *Fix $h \in [H]$ and $\delta \in (0,1)$, $\rho \geq 0$. Suppose that Assumption 1 and Assumption 2 holds. Then with probability at least $1-\delta$, the following holds:*

*(1) For each $t \in [T]$,*

$$\mathcal{E}_{\mathcal{D}^{(t)},h}(Q_h^\sharp, Q_{h+1}^\sharp; R^\sharp) - \inf_{g_h \in \mathcal{G}_h} \mathcal{E}_{\mathcal{D}^{(t)},h}(g_h, Q_{h+1}^\sharp; R^\sharp) \leq O\big(TH\varepsilon_{\mathrm{app}}^2 + TH\rho + \log(N(\rho)/\delta)\big).$$

*(2) For each $t \in [T]$, for all $f_h \in \mathcal{F}_h$, $f_{h+1} \in \mathcal{F}_{h+1}$, and $R_h \in \mathcal{R}_h$,*

$$\frac{1}{2}\sum_{k=1}^{t} \mathbb{E}^{\pi^{(k)}}\big(f_h(s_h, a_h) - [\mathcal{T}_{R,h}f_{h+1}](s_h, a_h)\big)^2$$

$$\leq \mathcal{E}_{\mathcal{D}^{(t)},h}(f_h, f_{h+1}; R_h) - \inf_{g_h \in \mathcal{G}_h} \mathcal{E}_{\mathcal{D}^{(t)},h}(g_h, f_{h+1}; R_h) + O\big(TH\varepsilon_{\mathrm{app}}^2 + TH\rho + \log(N(\rho)/\delta)\big),$$

*where we use $O(\cdot)$ to hide absolute constant for simplicity.*

**Proof of Lemma 17.** Fix $h \in [H]$ and denote $N := N(\rho)$. We let $\mathcal{F}_{h+1}'$ be a minimal $\rho$-covering of $\mathcal{F}_{h+1}$, and let $\mathcal{R}_h'$ be a minimal $\rho$-covering of $\mathcal{R}_h$. By definition, $|\mathcal{F}_{h+1}'| \leq N, |\mathcal{R}_h'| \leq N$.

In the following, we adopt the notation of the proof of Lemma 16. Recall that conditional on $\mathcal{D}^{(t-1)}$,

$$\tau^{(t,\ell)} = (s_1^{(t,\ell)}, a_1^{(t,\ell)}, \cdots, s_H^{(t,\ell)}, a_H^{(t,\ell)}) \sim \pi^{(t)} \circ_\ell \pi_{\mathrm{ref}},$$

and $\tau^{(t,1)}, \cdots, \tau^{(t,H)}$ are independent conditional on $\mathcal{D}^{(t-1)}$. For simplicity, we denote $x^{(t,\ell)} := (s_h^{(t,\ell)}, a_h^{(t,\ell)})$.

Fix $f_{h+1} \in \mathcal{F}_{h+1}' \cup \{Q_{h+1}^\sharp\}$ and $R_h \in \mathcal{R}_h' \cup \{R_h^\sharp\}$, we consider

$$y^{(t,\ell)} := f_{h+1}(s_{h+1}^{(t,\ell)}) + R_h(s_h^{(t,\ell)}, a_h^{(t,\ell)}).$$

and it holds that

$$\mathbb{E}\big[y^{(t,\ell)}\big|\mathcal{D}^{(t-1)}, x^{(t,\ell)}\big] = [\mathcal{T}_{R,h}f_{h+1}](x^{(t,\ell)}).$$

Then, for any $g_h \in \mathcal{G}_h$, it holds that

$$\mathcal{E}_{\mathcal{D}^{(t)},h}(g_h, f_{h+1}; R_h) = \sum_{k=1}^{t}\sum_{\ell=1}^{H}(g_h(x^{(t,\ell)}) - y^{(t,\ell)})^2,$$

and we also have

$$\sum_{k=1}^{t}\sum_{\ell=1}^{H}\mathbb{E}^{\pi^{(k)}\circ_\ell \pi_{\mathrm{ref}}}(g_h(s_h, a_h) - [\mathcal{T}_{R,h}f_{h+1}](s_h, a_h))^2 = \sum_{k=1}^{t}\sum_{\ell=1}^{H}\mathbb{E}\left[(g_h(x^{(k,\ell)}) - [\mathcal{T}_{R,h}f_{h+1}](x^{(k,\ell)}))^2\Big|\mathcal{D}^{(k-1)}\right]$$

Then, applying Proposition 7 with the function class $\mathcal{H} = \mathcal{G}_h$ yields that with probability at least $1 - \frac{\delta}{2N}$, the following holds:

(a) For each $t \in [T]$, for any $g_h \in \mathcal{G}_h$,

$$\frac{1}{2}\sum_{k=1}^{t}\mathbb{E}^{\pi^{(k)}}(g_h(s_h, a_h) - [\mathcal{T}_{R,h}f_{h+1}](s_h, a_h))^2$$

$$\leq \mathcal{E}_{\mathcal{D}^{(t)},h}(g_h, f_{h+1}; R_h) - \inf_{g_h' \in \mathcal{G}_h}\mathcal{E}_{\mathcal{D}^{(t)},h}(g_h', f_{h+1}; R_h) + O\big(\log(N/\delta) + TH\varepsilon_{\mathrm{app}}^2 + TH\rho\big).$$

(b) When $f_{h+1} = Q_{h+1}^\sharp$ and $R_h = R_h^\sharp$, the function $Q_h^\sharp \in \mathcal{F}_h \subseteq \mathcal{G}_h$ satisfies the inequality $\|Q_h^\sharp - \mathcal{T}_{R^\sharp,h}Q_{h+1}^\sharp\|_\infty \leq 3\varepsilon_{\mathrm{app}}$, and thus

$$\mathcal{E}_{\mathcal{D}^{(t)},h}(Q_h^\sharp, Q_{h+1}^\sharp; R_h^\sharp) \leq \inf_{g_h \in \mathcal{G}_h}\mathcal{E}_{\mathcal{D}^{(t)},h}(g_h, Q_{h+1}^\sharp; R_h^\sharp) + O\big(\log(N/\delta) + TH\varepsilon_{\mathrm{app}}^2 + TH\rho\big).$$

Therefore, taking the union bound, we know that the inequalities (a) and (b) above hold simultaneously with probability at least $1 - \delta$ for all $f_{h+1} \in \mathcal{F}'_{h+1} \cup \{Q^\sharp_{h+1}\}$ and $R_h \in \mathcal{R}'_h \cup \{R^\sharp_h\}$. In particular, we have completed the proof of (1).

To prove (2), we only need to note that $\mathcal{G}_h \subseteq \mathcal{F}_h$, and for any $f_{h+1} \in \mathcal{F}_{h+1}, R_h \in \mathcal{R}_h$, there exists $f'_{h+1} \in \mathcal{F}'_{h+1}, R'_h \in \mathcal{R}'_h$ such that $\|f_{h+1} - f'_{h+1}\|_\infty \leq \rho, \|R_h - R'_h\|_\infty \leq \rho$. Therefore, by the standard covering argument and the fact that $\pi^{(k)} \circ_H \pi_{\mathrm{ref}} = \pi^{(k)}$, we have also shown (2). $\qquad\square$

# D   Proof of Theorem 2

In this section, we provide the proof of Theorem 2, which is a direct adaption of the proof of Theorem 1 in Appendix C. We first present a more detailed statement of the upper bound (with any parameter $\lambda > 0$).

**Theorem 18.** *Suppose that Assumption 1 holds. Then with probability at least $1 - \delta$, Algorithm 2 achieves*

$$\frac{1}{T} \sum_{t=1}^T \left( V^\star(s_1^{(t)}) - V^{\pi^{(t)}}(s_1^{(t)}) \right) \leq \varepsilon_{\mathrm{app}} + O(1) \cdot \left[ \frac{H^3 \log(N_{\mathcal{F},T}/\delta) + T\varepsilon^2_{\mathrm{app}}}{\lambda} + \frac{\lambda H C'_{\mathrm{cov}}(\Pi)}{T} \right],$$

We also work with a slightly relaxed version of Assumption 3.

**Assumption 4.** *Under any policy $\pi$, for each $h \in [H]$, it holds that almost surely*
$$Q^\star_h(s_h, a_h) = R^\star_h(s_h, a_h) + V^\star_{h+1}(s_{h+1}).$$

Further, to simply the notation, for each $f \in \mathcal{F}$, we recall that the induced reward model $R^f$ is defined as $R^f_1(s,a) := f_1(s,a), R^f_h(s,a) = f_h(s,a) - f_h(s)$, which implies

$$R^f(\tau) = \sum_{h=1}^H f_h(s_h, a_h) - f_{h+1}(s_{h+1}).$$

**Uniform convergence.**   For each $t \in [T]$, we define $\mathcal{D}^{(t-1)} := \{(\tau^{(k)}, r^{(k)})\}_{k<t}$ be the data collected before $t$th iteration. We also recall that by definition (8), we have

$$\mathcal{L}^{\mathsf{BR}}_{\mathcal{D}^{(t)}}(f) := \sum_{k=1}^t \left( \sum_{h=1}^H \left[ f_h(s_h^{(k)}, a_h^{(k)}) - f_{h+1}(s_{h+1}^{(k)}) \right] - r^{(k)} \right)^2 = \sum_{k=1}^t \left( R^f(\tau^{(k)}) - r^{(k)} \right)^2.$$

Therefore, a direct instantiation of Proposition 7 on the class $\mathcal{R} := \{R^f : f \in \mathcal{F}\}$ yields the following proposition.

**Proposition 19.** *Let $\delta \in (0,1), \rho \geq 0$. Suppose that Assumption 1 and Assumption 4 holds. Then with probability at least $1 - \delta$, for all $t \in [T], f \in \mathcal{F}$, it holds that*

$$\frac{1}{2} \sum_{k=1}^t \mathbb{E}^{\pi^{(k)}} \left( R^f(\tau) - R^\star(\tau) \right)^2 \leq \mathcal{L}^{\mathsf{BR}}_{\mathcal{D}^{(t)}}(f) - \mathcal{L}^{\mathsf{BE}}_{\mathcal{D}^{(t)}}(Q^\sharp) + \kappa,$$

*where*

$$\kappa = C \left( H^3 \log(N_{\mathcal{F}}(\alpha)/\delta) + TH\alpha + T\varepsilon^2_{\mathrm{app}} \right),$$

*$C > 0$ is an absolute constant, and we denote $N_{\mathcal{F}}(\alpha) := \max_{h \in [H]} N(\mathcal{F}_h, \alpha)$ for any $\alpha \geq 0$.*

**Performance difference decomposition.**   In this setting, we can rewrite the decomposition (16) as

$$\begin{aligned}
f_1^{(t)}(s_1) - V^{\pi^{(t)}}(s_1) &= \sum_{h=1}^H \mathbb{E}^{\pi^{(t)}} \left[ f_h^{(t)}(s_h, a_h) - R^\star_h(s_h, a_h) - f_{h+1}^{(t)}(s_{h+1}) \right] \\
&= \mathbb{E}^{\pi^{(t)}} \left[ \sum_{h=1}^H \left[ f_h^{(t)}(s_h, a_h) - f_{h+1}^{(t)}(s_{h+1}) \right] - \sum_{h=1}^H R^\star_h(s_h, a_h) \right] \quad (21) \\
&= \mathbb{E}^{\pi^{(t)}} [R^{(t)}(\tau) - R^\star(\tau)],
\end{aligned}$$

where we denote $R^{(t)} := R^{f^{(t)}}$, which is a reward model given by
$$R_1^{(t)}(s,a) := f_1^{(t)}(s,a), \qquad R_h^{(t)}(s,a) = f_h^{(t)}(s,a) - f_h^{(t)}(s).$$

**Optimism.** Similar to Appendix C.1, we use the fact that from (9),

$$f^{(t)} = \max_{f \in \mathcal{F}} \lambda f_1(s_1^{(t)}) - \mathcal{L}_{\mathcal{D}^{(t-1)}}^{\mathsf{BR}}(f),$$

and hence

$$\lambda f_1^{(t)}(s_1^{(t)}) - \mathcal{L}_{\mathcal{D}^{(t-1)}}^{\mathsf{BR}}(f^{(t)}) \geq \lambda V_1^{\sharp}(s_1^{(t)}) - \mathcal{L}_{\mathcal{D}^{(t-1)}}^{\mathsf{BR}}(f^{(t)}).$$

Using $\left| V_1^{\sharp}(s_1^{(t)}) - V_1^{\star}(s_1^{(t)}) \right| \leq \varepsilon_{\mathrm{app}}$, (21) and Proposition 19, we now deduce that

$$V^{\star}(s_1^{(t)}) - V^{\pi^{(t)}}(s_1^{(t)}) \leq \varepsilon_{\mathrm{app}} + \mathbb{E}^{\pi^{(t)}}[R^{(t)}(\tau) - R^{\star}(\tau)] - \frac{\mathcal{L}_{\mathcal{D}^{(t-1)}}^{\mathsf{BR}}(f^{(t)}) - \mathcal{L}_{\mathcal{D}^{(t-1)}}^{\mathsf{BE}}(Q^{\sharp})}{\lambda}$$

$$\leq \varepsilon_{\mathrm{app}} + \frac{\kappa}{\lambda} + \mathbb{E}^{\pi^{(t)}}[R^{(t)}(\tau) - R^{\star}(\tau)] - \frac{1}{2\lambda} \sum_{k=1}^{t-1} \mathbb{E}^{\pi^{(k)}}(R^{(t)}(\tau) - R^{\star}(\tau))^2.$$

$$(22)$$

Therefore, it remains to prove an analogue to Proposition 13.

**Coverability argument.** We strength Proposition 13 using the deterministic nature of the underlying MDP. For each $s \in \mathcal{S}$ and $h \in [H]$, we define

$$\mathcal{S}_h(s; \Pi) := \{(s', a) : \exists \pi \in \Pi, \text{ under } \pi \text{ and } s_1 = s, \text{ it holds that } s_h = s', a_h = a\},$$

and $N_h(s; \Pi) := |\mathcal{S}_h(s; \Pi)|$.

**Proposition 20.** *Let $B \geq 1$. For any initial state $s_1 \in \mathcal{S}$, any sequence of reward functions $R^{(1)}, \cdots, R^{(T)}$ and any sequence of policies $\pi^{(1)}, \cdots, \pi^{(T)}$, it holds that*

$$\sum_{t=1}^{T} \mathbb{E}^{\pi^{(t)}} \left[ R^{(t)}(\tau) - R^{\star}(\tau) \middle| s_1 \right]$$

$$\leq \sqrt{2N(s_1) \log\left(1 + \frac{4TH}{B}\right) \cdot \left[ 2TB + \sum_{1 \leq k < t \leq T} \mathbb{E}^{\pi^{(k)}} \left[ (R^{(t)}(\tau) - R^{\star}(\tau))^2 \middle| s_1 \right] \right]},$$

*where $N(s_1) := \sum_{h=1}^{H} N_h(s_1; \Pi)$, and the conditional distribution $\mathbb{E}^{\pi^{(t)}} [\cdot | s_1]$ is taken over the expectation of $\tau$ generated by executing policy $\pi$ starting with the initial state $s_1$.*

The proof of Proposition 20 is deferred to the end of this section.

**Finalizing the proof.** With the above preparation, we now finalize the proof of Theorem 2. Taking summation of (22) over $t = 1, 2, \cdots, T$, we have

$$\sum_{t=1}^{T} V^{\star}(s_1^{(t)}) - V^{\pi^{(t)}}(s_1^{(t)})$$

$$\leq T\varepsilon_{\mathrm{app}} + \frac{T\kappa}{\lambda} + \sum_{t=1}^{T} \mathbb{E}^{\pi^{(t)}}[R^{(t)}(\tau) - R^{\star}(\tau)] - \frac{1}{2\lambda} \sum_{1 \leq k < t \leq T} \mathbb{E}^{\pi^{(k)}}(R^{(t)}(\tau) - R^{\star}(\tau))^2$$

$$= T\varepsilon_{\mathrm{app}} + \frac{T\kappa}{\lambda} + \mathbb{E}_{s_1 \sim \rho}\left[ \sum_{t=1}^{T} \mathbb{E}^{\pi^{(t)}} \left[ R^{(t)}(\tau) - R^{\star}(\tau) \middle| s_1 \right] - \frac{1}{2\lambda} \sum_{1 \leq k < t \leq T} \mathbb{E}^{\pi^{(k)}} \left[ (R^{(t)}(\tau) - R^{\star}(\tau))^2 \middle| s_1 \right] \right]$$

$$\leq T\varepsilon_{\mathrm{app}} + \frac{2T\kappa}{\lambda} + \mathbb{E}_{s_1 \sim \rho}\left[ N(s_1) \lambda \log\left(1 + \frac{TH}{\kappa}\right) \right],$$

where the last inequality follows from Proposition 20 and Cauchy inequality. This is the desired upper bound. $\qquad\square$

**Proof of Proposition 20.** In the following proof, we assume $s_1 \in \mathcal{S}$ is fixed. Consider

$$\mathcal{I} := \{(h, s, a) : h \in [H], (s, a) \in \mathcal{S}_h(s_1; \Pi)\} \subseteq [H] \times \mathcal{S} \times \mathcal{A}.$$

Note that $|\mathcal{I}| = \sum_{h=1}^{H} N_h(s_1; \Pi) = N(s_1)$. By definition, for any policy $\pi$, there is a unique pair $(s_h^\pi, a_h^\pi) \in \mathcal{S}_h(s_1; \Pi)$, such that under $\pi$ and starting from $s_1$, we have $s_h = s_h^\pi, a_h = a_h^\pi$ deterministically.

For each $t \in [T]$, we consider the following vectors indexed by $\mathcal{I}$:

$$\psi^{(t)} := \left[R_h^{(t)}(s, a) - R_h^\star(s, a)\right]_{(h,s,a) \in \mathcal{I}} \in \mathbb{R}^{\mathcal{I}},$$

$$\phi^{(t)} := \left[\mathbb{P}^{\pi^{(t)}}(s_h = s, a_h = a | s_1)\right]_{(h,s,a) \in \mathcal{I}} = \sum_{h=1}^{H} e_{(h, s_h^{\pi^{(t)}}, a_h^{\pi^{(t)}})} \in \mathbb{R}^{\mathcal{I}}.$$

With this definition, it holds that for any $k, t \in [T]$,

$$\mathbb{E}^{\pi^{(k)}}\left[R^{(t)}(\tau) - R^\star(\tau) | s_1\right] = \sum_{h=1}^{H} \left[R^{(t)}(s_h^{\pi^{(k)}}, a_h^{\pi^{(k)}}) - R^\star(s_h^{\pi^{(k)}}, a_h^{\pi^{(k)}})\right] = \langle \phi^{(k)}, \psi^{(t)} \rangle.$$

Therefore, we apply the elliptical potential argument (Lattimore and Szepesvári, 2020). Let $V_t := \sum_{k<t} \phi^{(k)}(\phi^{(k)})^\top + B\mathbf{I}$. Then it holds that

$$\sum_{t=1}^{T} |\langle \phi^{(t)}, \psi^{(t)} \rangle| \leq \sum_{t=1}^{T} \min\{\|\phi^{(t)}\|_{V_t^{-1}}, 1\} \cdot \max\{\|\psi^{(t)}\|_{V_t}, 1\}$$

$$\leq \sqrt{\sum_{t=1}^{T} \min\{\|\phi^{(t)}\|_{V_t^{-1}}^2, 1\}} \cdot \sqrt{\sum_{t=1}^{T} \max\{\|\psi^{(t)}\|_{V_t}^2, 1\}}.$$

Note that

$$\sum_{t=1}^{T} \max\{\|\psi^{(t)}\|_{V_t}^2, 1\} \leq \sum_{t=1}^{T} \left[1 + B\|\psi^{(t)}\|^2 + \sum_{k=1}^{t-1} \langle \phi^{(k)}, \psi^{(t)} \rangle^2\right]$$

$$\leq T(1 + 4B|\mathcal{I}|) + \sum_{1 \leq k < t \leq T} \mathbb{E}^{\pi^{(k)}}\left[\left(R^{(t)}(\tau) - R^\star(\tau)\right)^2 \Big| s_1\right],$$

and by Lattimore and Szepesvári (2020), we have

$$\sum_{t=1}^{T} \min\{\|\phi^{(t)}\|_{V_t^{-1}}^2, 1\} \leq 2|\mathcal{I}| \log\left(1 + \frac{TH}{|\mathcal{I}|B}\right).$$

Combining the inequalities above and rescale $B \leftarrow \frac{B}{4|\mathcal{I}|}$ completes the proof. $\qquad\square$

# E  Proofs from Section 4

We present the full description of our algorithm or preference-based RL as follows.

### E.1  Proof of Theorem 3

For each $t \in [T]$, we write $\mathcal{D}^{(t)}$ to be the dataset maintained by Algorithm 1 at the end of the $t$th iteration, i.e.,

$$\mathcal{D}^{(t)} = \{(\tau^{(k,h,+)}, \tau^{(k,h,-)}, y^{(k,h)})\}_{k \leq t, h \in [H]}.$$

Note that for each $t \in [T]$, $h \in [H]$, we have $\pi^{(t,h,-)} = \pi_{\text{ref}}$. Therefore, for each $R \in \mathcal{R}$, we define $V_R^{\text{ref}} := \mathbb{E}^{\pi_{\text{ref}}}[R(\tau)]$ and recall that

$$\widehat{V}_{\mathcal{D},R}^{\text{ref}} := \frac{1}{|\mathcal{D}|} \sum_{(\tau^+, \tau^-, y) \in \mathcal{D}} R(\tau^-).$$

The following lemma follows from the standard uniform convergence rate with Hoeffding's inequality and the union bound.

**Algorithm 3** Outcome-Based Exploration for Preference-based RL

**input:** Function class $\mathcal{F}$, parameter $\lambda > 0$, reference policy $\pi_{\text{ref}}$.
**initialize:** $\mathcal{D} \leftarrow \emptyset$.

1: **for** $t = 1, 2, \ldots, T$ **do**
2:      Compute the optimistic estimates through (12):

$$(f^{(t)}, R^{(t)}) = \max_{f \in \mathcal{F}, R \in \mathcal{R}} \lambda \Big[ f_1(s_1) - \widehat{V}_{\mathcal{D},R}^{\text{ref}} \Big] - \mathcal{L}_{\mathcal{D}}^{\text{BE}}(f; R) - \mathcal{L}_{\mathcal{D}}^{\text{PbRM}}(R),$$

3:      Select policy $\pi^{(t)} \leftarrow \pi_{f^{(t)}}$.
4:      **for** $h = 1, 2, \cdots, H$ **do**
5:          Execute $\pi^{(t)} \circ_h \pi_{\text{ref}}$ for two episode and obtain two trajectories $(\tau^{(t,h,+)}, \tau^{(t,h,-)})$ and preference feedback $y^{(t,h)}$.
6:
7:          Update dataset: $\mathcal{D} \leftarrow \mathcal{D} \cup \{(\tau^{(t,h,+)}, \tau^{(t,h,-)}, y^{(t,h)})\}$.
8:      **end for**
9: **end for**
10: Output $\widehat{\pi} = \text{Unif}(\pi^{(1:T)})$.

---

**Lemma 21.** *Let $\delta \in (0,1), \rho \geq 0$. Suppose that Assumption 1 and Assumption 2 holds. Then with probability at least $1 - \delta$, for all $t \in [T], R \in \mathcal{R}$, it holds that*

$$\left| \widehat{V}_{\mathcal{D}^{(t)},R}^{\text{ref}} - V_R^{\text{ref}} \right| \leq \sqrt{\frac{\log(2TN(\rho)/\delta)}{t}} + H\rho.$$

We summarize the uniform concentration results for the loss $\mathcal{L}_{\mathcal{D}^{(t)}}^{\text{BE}}$ and $\mathcal{L}_{\mathcal{D}^{(t)}}^{\text{PbRM}}$ as follows. The proof is analogous to Proposition 11 and is provided in Appendix E.2.

**Proposition 22.** *Let $\delta \in (0,1), \rho \geq 0$. Suppose that Assumption 1 and Assumption 2 holds. Then with probability at least $1 - \delta$, for all $t \in [T], f \in \mathcal{F}, R \in \mathcal{R}$, it holds that*

$$\left| \widehat{V}_{\mathcal{D}^{(t)},R}^{\text{ref}} - V_R^{\text{ref}} \right| \leq \sqrt{\frac{\kappa}{tH}},$$

$$\sum_{k \leq t} \sum_{h=1}^{H} \mathbb{E}^{\pi^{(k,h,+)}, \pi^{(k,h,-)}} \left( [R(\tau^+) - R(\tau^-)] - [R^\star(\tau^+) - R^\star(\tau^-)] \right)^2 \leq C_\beta \left[ \mathcal{L}_{\mathcal{D}^{(t)}}^{\text{PbRM}}(R) - \mathcal{L}_{\mathcal{D}^{(t)}}^{\text{PbRM}}(R^\sharp) \right] + C_\beta H\kappa,$$

$$\sum_{k \leq t} \sum_{h=1}^{H} \mathbb{E}^{\pi^{(k)}} \left( f_h(s_h, a_h) - [\mathcal{T}_{R,h} f_{h+1}](s_h, a_h) \right)^2 \leq 2 \left[ \mathcal{L}_{\mathcal{D}^{(t)}}^{\text{BE}}(f; R) - \mathcal{L}_{\mathcal{D}^{(t)}}^{\text{BE}}(Q^\sharp; R^\sharp) \right] + H\kappa,$$

*where $C_\beta = \frac{4e^{2\beta}}{\beta^2}$,*

$$\kappa = C \left( \log N(\rho) + \log(TH/\delta) + TH^2(\beta + 1)(\varepsilon_{\text{app}}^2 + \rho) \right),$$

*and $C > 0$ is an absolute constant.*

In the following, we condition on the success event of Proposition 22. Note that $\pi^{(t,h,-)} \equiv \pi_{\text{ref}}$, and hence Proposition 22 implies that for all $R \in \mathcal{R}, t \in [T]$,

$$\sum_{k \leq t} \sum_{h=1}^{H} \mathbb{E}^{\pi^{(k)} \circ_h \pi_{\text{ref}}} \left( [R(\tau) - R^\star(\tau)] - [V_R^{\text{ref}} - V_{R^\star}^{\text{ref}}] \right)^2 \leq C_\beta \left[ \mathcal{L}_{\mathcal{D}^{(t)}}^{\text{PbRM}}(R) - \mathcal{L}_{\mathcal{D}^{(t)}}^{\text{PbRM}}(R^\sharp) \right] + C_\beta H\kappa.$$

Therefore, for any reward function $R$, we define $\widetilde{R}$ as $\widetilde{R}_1(s,a) = R_1(s,a) - V_R^{\text{ref}}$ and $\widetilde{R}_h(s,a) = R_h(s,a)$ for $h > 1$. Then it is clear that $\widetilde{R}(\tau) = R(\tau) - V_R^{\text{ref}}$, and for all $R \in \mathcal{R}, t \in [T]$, we have

$$\sum_{k \leq t} \sum_{h=1}^{H} \mathbb{E}^{\pi^{(k)} \circ_h \pi_{\text{ref}}} \left( \widetilde{R}(\tau) - \widetilde{R}^\star(\tau) \right)^2 \leq C_\beta \left[ \mathcal{L}_{\mathcal{D}^{(t)}}^{\text{PbRM}}(R) - \mathcal{L}_{\mathcal{D}^{(t)}}^{\text{PbRM}}(R^\sharp) \right] + C_\beta H\kappa. \quad (23)$$

**Performance difference decomposition.** In this setting, we re-write (17) as follows:

$$f_1^{(t)}(s_1) - V^{\pi^{(t)}}(s_1) = \sum_{h=1}^{H} \mathbb{E}^{\pi^{(t)}} \left[ f_h^{(t)}(s_h, a_h) - [\mathcal{T}_{R^{(t)}} f_{h+1}^{(t)}](s_h, a_h) \right]$$

$$+ \mathbb{E}^{\pi^{(t)}} \left[ \sum_{h=1}^{H} R_h^{(t)}(s_h, a_h) - \sum_{h=1}^{H} R_h^{\star}(s_h, a_h) \right]$$

$$= \sum_{h=1}^{H} \mathbb{E}^{\pi^{(t)}} e_h^{(t)}(s_h, a_h) + \mathbb{E}^{\pi^{(t)}} \left[ \widetilde{R}^{(t)}(\tau) - \widetilde{R}^{\star}(\tau) \right] + V_{R^{(t)}}^{\text{ref}} - V_{R^{\star}}^{\text{ref}},$$

where we recall that we denote $e_h^{(t)} := f_h^{(t)} - \mathcal{T}_{R^{(t)}} f_{h+1}^{(t)}$. Therefore, we re-organize the equality as

$$\left[ f_1^{(t)}(s_1) - V_{R^{(t)}}^{\text{ref}} \right] - \left[ V^{\pi^{(t)}}(s_1) - V_{R^{\star}}^{\text{ref}} \right] = \mathbb{E}^{\pi^{(t)}} \left[ \widetilde{R}^{(t)}(\tau) - \widetilde{R}^{\star}(\tau) \right] + \sum_{h=1}^{H} \mathbb{E}^{\pi^{(t)}} e_h^{(t)}(s_h, a_h). \tag{24}$$

With the above preparation, we present the proof of Theorem 3, which closely follows the proof of Theorem 1 in Appendix C.1.

**Proof of Theorem 3.** By definition, for each $t \in [T]$,

$$(f^{(t)}, R^{(t)}) = \max_{f \in \mathcal{F}, R \in \mathcal{R}} \lambda \left[ f_1(s_1) - \widehat{V}_{\mathcal{D}^{(t-1)}, R}^{\text{ref}} \right] - \mathcal{L}_{\mathcal{D}^{(t-1)}}^{\text{BE}}(f; R) - \mathcal{L}_{\mathcal{D}^{(t-1)}}^{\text{PbRM}}(R).$$

Therefore, using $Q^{\sharp} \in \mathcal{F}, R^{\sharp} \in \mathcal{R}$, we have

$$\left[ f_1^{(t)}(s_1) - \widehat{V}_{\mathcal{D}^{(t-1)}, R^{(t)}}^{\text{ref}} \right] - \left[ V_1^{\sharp}(s_1) - \widehat{V}_{\mathcal{D}^{(t-1)}, R^{\sharp}}^{\text{ref}} \right]$$

$$\leq -\frac{\mathcal{L}_{\mathcal{D}^{(t-1)}}^{\text{BE}}(f^{(t)}; R^{(t)}) - \mathcal{L}_{\mathcal{D}^{(t-1)}}^{\text{BE}}(Q^{\sharp}; R^{\sharp})}{\lambda} - \frac{\mathcal{L}_{\mathcal{D}^{(t-1)}}^{\text{PbRM}}(R^{(t)}) - \mathcal{L}_{\mathcal{D}^{(t-1)}}^{\text{PbRM}}(R^{\sharp})}{\lambda}.$$

Using the decomposition (24), Proposition 22, and the fact that $\left| V_1^{\sharp}(s_1) - V_1^{\star}(s_1) \right| \leq \varepsilon_{\text{app}}$, $\left| R^{\sharp}(\tau) - R^{\star}(\tau) \right| \leq H \varepsilon_{\text{app}}$, we have

$$V_1^{\star}(s_1) - V^{\pi^{(t)}}(s_1) \leq (H+1)\varepsilon_{\text{app}} + \left| V_{R^{(t)}}^{\text{ref}} - \widehat{V}_{\mathcal{D}^{(t-1)}, R^{(t)}}^{\text{ref}} \right| + \left| V_{R^{\sharp}}^{\text{ref}} - \widehat{V}_{\mathcal{D}^{(t-1)}, R^{\sharp}}^{\text{ref}} \right| + \frac{2H\kappa}{\lambda}$$

$$+ \sum_{h=1}^{H} \left( \mathbb{E}^{\pi^{(t)}} \left[ e_h^{(t)}(s_h, a_h) \right] - \frac{1}{C_{\beta}\lambda} \sum_{k<t} \mathbb{E}^{\pi^{(k)}} e_h^{(t)}(s_h, a_h)^2 \right)$$

$$+ \mathbb{E}^{\pi^{(t)}} \left[ \widetilde{R}^{(t)}(\tau) - \widetilde{R}^{\star}(\tau) \right] - \frac{1}{2\lambda} \sum_{k<t} \sum_{h=1}^{H} \mathbb{E}^{\pi^{(k)} \circ_h \pi_{\text{ref}}} \left( \widetilde{R}^{(t)}(\tau) - \widetilde{R}^{\star}(\tau) \right)^2. \tag{25}$$

Taking summation over $t = 1, 2, \cdots, T$ and apply Proposition 12, Proposition 13, and Lemma 21 yields

$$\sum_{t=1}^{T} V_1^{\star}(s_1) - V^{\pi^{(t)}}(s_1) \leq O(1) \cdot \left[ H(\varepsilon_{\text{app}} + \rho) + \sqrt{T\kappa} + \frac{TH\kappa}{\lambda} + C_{\beta}\lambda H C_{\text{cov}} \log \left( 1 + \frac{C_{\text{cov}}T}{\kappa} \right) \right].$$

This is the desired upper bound. $\square$

### E.2 Proof of Proposition 22

The inequality involving $\mathcal{L}_{\mathcal{D}^{(t)}}^{\text{BE}}$ is implied by Proposition 11 and proven in Appendix C.3. In the following, we only need to prove the inequality involving $\mathcal{L}_{\mathcal{D}^{(t)}}^{\text{PbRM}}$ by invoking Proposition 8.

Consider the class $\Theta = \mathcal{R} \cup \{R^{\star}\}$, $\mathcal{X} = (\mathcal{S} \times \mathcal{A})^H \times (\mathcal{S} \times \mathcal{A})^H$, and $\mathcal{Y} = \{0, 1\}$. For any $R \in \Theta$, we define

$$P_R(1|\tau^+, \tau^-) = \frac{\exp\left(\beta R(\tau^+)\right)}{\exp\left(\beta R(\tau^+)\right) + \exp\left(\beta R(\tau^-)\right)}, \quad P_R(0|\tau^+, \tau^-) = \frac{\exp\left(\beta R(\tau^-)\right)}{\exp\left(\beta R(\tau^+)\right) + \exp\left(\beta R(\tau^-)\right)},$$

following Definition 3.

Recall that $\mathcal{D}^{(t-1)} = \{(\tau^{(k,h,+)}, \tau^{(k,h,-)}, y^{(k,h)})\}_{k<t,h\in[H]}$ is the history up to the $t$-th iteration. For simplicity, we denote $x^{(t,h)} := (\tau^{(t,h,+)}, \tau^{(t,h,-)})$. Note that $(x^{(t,1)}, y^{(t,1)}), \cdots, (x^{(t,H)}, y^{(t,H)})$. Then it is clear that for all $t \in [T], h \in [H]$,

$$\mathbb{P}(y^{(t,h)}|x^{(t,h)}, \mathcal{D}^{(t-1)}) = P_{R^\star}(y^{(t,h)}|x^{(t,h)}),$$

and it also holds that

$$L(R(\tau^+) - R(\tau^-), y) = -\log P_R(y|\tau^+, \tau^-), \qquad \forall y \in \{0,1\}.$$

Further, noting that $N_{\log}(\Theta, 2H\beta\rho) \leq N(\mathcal{R}, \rho) + 1$. Therefore, applying Proposition 8 gives the following result: with probability at least $1 - \frac{\delta}{2}$, for any $R \in \mathcal{R}, t \in [T]$,

$$\sum_{k=1}^{t} \sum_{h=1}^{H} \mathbb{E}^{\pi^{(k,h,+)}, \pi^{(k,h,-)}} D_{\mathrm{H}}^2\big(P_R(\cdot|\tau^+, \tau^-), P_{R^\star}(\cdot|\tau^+, \tau^-)\big)$$

$$\leq \frac{1}{2} \sum_{(\tau^+, \tau^-, y)} \big[L(R(\tau^+) - R(\tau^-), y) - L(R^\star(\tau^+) - R^\star(\tau^-), y)\big]$$

$$+ \log(N(\mathcal{R}, H\rho) + 1) + \log(2/\delta) + TH^3\beta\rho$$

$$\leq \frac{1}{2}\big[\mathcal{L}_{\mathcal{D}^{(t)}}^{\mathsf{PbRM}}(R) - \mathcal{L}_{\mathcal{D}^{(t)}}^{\mathsf{PbRM}}(R^\sharp)\big] + \frac{1}{2}H\kappa,$$

where the second inequality uses the fact that $\big|R^\sharp(\tau) - R^\star(\tau)\big| \leq H\varepsilon_{\mathrm{app}}$. Finally, note that $D_{\mathrm{H}}^2(\mathrm{Bern}(p), \mathrm{Bern}(q)) \geq \frac{1}{2}(p-q)^2$ and

$$\left|\frac{1}{e^{\beta w} + 1} - \frac{1}{e^{\beta w'} + 1}\right| \geq \frac{\beta}{2e^\beta}|w - w'|, \qquad \forall w, w' \in [-1, 1].$$

Therefore, using the definition of $P_R$ completes the proof. $\qquad\square$

# F Proofs of Lower Bounds

## F.1 Hard Case of Learning with Fitted Reward Models

As mentioned in Section 3.1, in Algorithm 1 the learner has to optimize over the reward class and value function class jointly. In the following, we argue that if the learner first learns a fitted reward model in the reward class, then optimizes the value function with the fitted rewards, the output policies at each iteration never converge to the optimal policy.

In detail, we consider algorithms in the form of Algorithm 4, where the learner fits the reward model $R^{(t)}$ at iteration $t$ first, then the learner calls algorithm alg, which takes per-step rewards data as input and outputs a policy $\pi_t$ at each iteration. To align with the structure of Algorithm 1, we take alg to be a single iteration of the GOLF algorithm in Jin et al. (2021a), i.e. $\pi^{(t)} = \pi_{f^{(t)}}$ where $f^{(t)} = \arg\max_{f \in \mathcal{F}^{(t)}} f(x_1, \pi_f(x_1))$. Here the confidence set $\mathcal{F}^{(t)}$ is defined as

$$\mathcal{F}^{(t)} = \left\{f \in \mathcal{F} : \mathcal{L}_{\mathcal{D}^{(t-1)}}^{\mathsf{BE}}(f; \widehat{R}^{(t-1)}) \leq \beta\right\}$$

with $\mathcal{L}_{\mathcal{D}}^{\mathsf{BE}}$ defined in Eq. (5).

Then we have the following proposition, which shows that this approach outputs suboptimal policies at every iteration in some special hard cases.

**Proposition 23.** *Consider Algorithm 4 with* alg *to a single iteration of the GOLF algorithm. After running $T$ iterations, the learner averages over all policies to output a policy. There exists an MDP class that realizes the ground truth MDP, such that the above algorithm outputs a policy which is at least $0.01$-suboptimal.*

***Proof of Proposition 23.*** We consider the following class of two-layer MDP, where $\mathcal{S}_1 = \{s_1\}$, $\mathcal{S}_2 = \{s_2\}$, and the action space to be $\mathcal{A} = \{a_1, a_2\}$. The transition models $\mathbb{T}$ are identical across the class, and have the following form:

$$\mathbb{T}(s_2 \mid s_1, a_i) = 1, \qquad \forall i \in \{1, 2\}.$$

---
**Algorithm 4** RL with fitted reward models
---
**input:** Algorithm alg, reward regression oracle $O$.

1: **Initialize** $\mathcal{D}_h^{(0)} = \emptyset$ for every $h \in [H]$
2: **for** $t = 1, 2, \ldots, T$ **do**
3:     Feed $\mathcal{D}^{(t-1)}$ to alg and receive $\pi^{(t)}$ from alg
4:     Execute $\pi^{(t)}$ and receive $(\tau^{(t)}, r^{(t)})$, where $\tau^{(t)} = (s_1^{(t)}, a_1^{(t)}, \cdots, s_H^{(t)}, a_H^{(t)})$
5:     Receive the fitted reward function from $O$:

$$\widehat{R}^{(t)} = \min_{R \in \mathcal{R}} \sum_{k=1}^{t} (R(\tau^{(k)}) - r^{(k)})^2.$$

6:     Let $\widehat{r}_h^{(t)} = \widehat{R}_h^{(t)}(s_h^{(t)}, a_h^{(t)})$ for each $h \in [H]$.
7:     Let $\mathcal{D}^{(t)} = \mathcal{D}^{(t-1)} \cup \{(s_1^{(t)}, a_1^{(t)}, \widehat{r}_1^{(t)}, \cdots, s_H^{(t)}, a_H^{(t)}, \widehat{r}_H^{(t)})\}$.
8: **end for**
---

The reward class is defined as $\mathcal{R} = \{R^1, R^2\}$, where

$$R^1(s_1, a_1) = R^1(s_1, a_2) = 0.20, \quad R^1(s_2, a_1) = 0.20, \quad R^1(s_2, a_2) = 0.19,$$
$$\text{and} \quad R^2(s_1, a_1) = R^2(s_1, a_2) = 0.00, \quad R^2(s_2, a_1) = 0.38, \quad R^2(s_2, a_2) = 0.39.$$

The $Q$-function class $\mathcal{Q}$ is defined as $\mathcal{Q} = \{Q^1, Q^2, Q^3, Q^4\}$, which takes value in Table 1 respectively. Notice that in all possible reward models and $Q$-functions, the values at $(s_1, a_1)$ and at $(s_1, a_2)$

Table 1: Value of $Q^1, Q^2, Q^3, Q^4$

|       | $(s_1, a_1)$ | $(s_1, a_2)$ | $(s_2, a_1)$ | $(s_2, a_2)$ |
|-------|------|------|------|------|
| $Q^1$ | 0.40 | 0.40 | 0.20 | 0.19 |
| $Q^2$ | 0.20 | 0.20 | 0.20 | 0.19 |
| $Q^3$ | 0.59 | 0.59 | 0.38 | 0.39 |
| $Q^4$ | 0.39 | 0.39 | 0.38 | 0.39 |

are the same. In the following, when without ambiguity we simply use $R(s_1)$ to denote $R(s_1, a_1)$ and $R(s_1, a_2)$, and use $Q(s_1)$ to denote $Q(s_1, a_1)$ and $Q(s_2, a_2)$.

We further suppose the ground truth model reward satisfies $R = R^1$, then we can verify that the optimal $Q$-function is $Q^1$. It is easy to verify that sets $\mathcal{Q}$ and $\mathcal{R}$ satisfy the completeness assumption. Hence, sets $\mathcal{Q}$ and $\mathcal{R}$ satisfy the realizability assumption Assumption 1 and the completeness assumption Assumption 2 with $\mathcal{G} = \mathcal{Q}$.

To see why this is a hard-case for GOLF type algorithms, we first notice that for any trajectory $\tau = (s_1, \tilde{a}_1, s_2, \tilde{a}_2)$ with outcome reward $r = R^1(s_1, \tilde{a}_1) + R^1(s_2, \tilde{a}_2)$ collected by the algorithm, we always have
$$r = R^2(s_1) + R^2(s_2, \tilde{a}_2).$$

Hence as long as $\mathcal{D}$ does not contain state-action pair $(s_2, a_1)$, when fitting the reward function using the following ERM oracle:
$$R = \underset{R \in \mathcal{R}}{\operatorname{argmin}} \sum_{(\tau, r) \in \mathcal{D}} (r(\tau) - r)^2,$$

the reward model $R^2$ always achieves the minimum. In the worst case, we assume the fitted reward models encountered by the learner at such rounds are always $R^2$.

In the following, we verify that by running the GOLF algorithm, the learner will not encounter the state-action pair $(s_2, a_1)$ at any round. We notice that the optimal policies of $Q^3$ and $Q^4$ all take $a_2$ at state $s_2$, and also that
$$Q^3(s_1) \geq Q^1(s_1) \quad \text{and} \quad Q^3(s_1) \geq Q^2(s_1).$$

Hence, to verify that the algorithm never chooses $a_1$ at state $s_2$, we only need to verify that if either $Q^1$ or $Q^2$ belongs to the confidence set, then $Q^3$ also belongs to the confidence set.

When the learner collects a new trajectory, two new pieces of data will be added to the dataset $\mathcal{D}$. If the trajectory does not pass through the state-action pair $(s_2, a_1)$, these two pieces of data will be in the following form:

$$(s_1, a_1, R^2(s_1)), \quad (s_2, a_2, R^2(s_2, a_2)) \quad \text{or} \quad (s_1, a_2, R^2(s_1)), \quad (s_2, a_2, R^2(s_2, a_2)).$$

No matter which one of these two, we have the following inequality for the sum of squared Bellman error across these two pieces of data

$$\mathcal{E}_1(Q^1)^2 + \mathcal{E}_2(Q^1)^2 = 0.20^2 + 0.20^2 \geq 0.20^2 = \mathcal{E}_1(Q^3)^2 + \mathcal{E}_2(Q^3)^2,$$
$$\mathcal{E}_1(Q^2)^2 + \mathcal{E}_2(Q^2)^2 = 0.01^2 + 0.28^2 \geq 0.20^2 = \mathcal{E}_1(Q^3)^2 + \mathcal{E}_2(Q^3)^2.$$

According to the construction of the confidence set, if either $Q^1$ or $Q^2$ belongs to the confidence set, then $Q^3$ belongs to the confidence set as well.

Therefore, no matter how many rounds the algorithm runs, the optimistic policy always takes action $a_2$ at state $s_2$. Hence the average policy $\widehat{\pi}$ also takes $a_2$ at $s_2$, which implies that

$$J(\pi^\star) - J(\widehat{\pi}) \geq 0.01.$$

$\square$

## F.2  Proof of Theorem 4

Fix a parameter $\varepsilon \in (0, 1)$ and $N \leq \left(\frac{1}{2\varepsilon}\right)^{d/2}$. Then, by the standard packing argument over sphere (see e.g., Li et al., 2022), there exists a set $\Theta = \{\theta_1, \cdots, \theta_N\} \subseteq \mathbb{S}^{d-1}$ such that

$$\|\theta_i - \theta_j\| \geq \sqrt{2\varepsilon}, \qquad \forall i \neq j.$$

This implies $\langle \theta_i, \theta_j \rangle \leq 1 - \varepsilon$ for any $i \neq j$.

**Construction.**   In the following, we set $b = 1 - \varepsilon$, and construct state space $\mathcal{S}$ as

$$\mathcal{S} = \mathcal{S}_1 \sqcup \mathcal{S}_2, \qquad \mathcal{S}_1 = \{s_1\}, \qquad \mathcal{S}_2 = \Theta,$$

and let action space $\mathcal{A} = \Theta$. The initial state is always $s_1$, and we define the transition $\mathbb{T}$ as

$$\mathbb{T}(s_2 = \theta \mid s_1, a = \theta) = 1, \qquad \forall \theta \in \Theta,$$

i.e., taking action $a = \theta$ at $s_1$ transits to $s_2 = \theta$ deterministically.

**Reward functions.**   For any $v \in \Theta$, we define the reward model $R^v$ as follows:

$$R_1^v(s, a) = \frac{1}{3}[\text{ReLU}(\langle a, v \rangle - b) + \langle a, v \rangle + 1] \in [0, 1], \qquad \forall a \in \Theta,$$
$$R_2^v(s, a) = \frac{1}{3}[1 - \langle s, v \rangle] \in [0, 1], \qquad \forall s \in \Theta.$$

Note that we can write $g_1(x) = \frac{1}{3}[\text{ReLU}(x - b) + x + 1], g_2(x) = \frac{1-x}{3}$, and then $g_2$ is a linear function, and

$$\frac{1}{3}|x - y| \leq |g_1(x) - g_2(y)| \leq \frac{2}{3}|x - y|, \qquad \forall x, y \in \mathbb{R}.$$

Hence, $g_1$ and $g_2$ are (well-conditioned) generalized linear functions, and hence $R_1^v$ and $R_2^v$ are both (well-conditioned) $d$-dimensional generalized linear functions.

We let $M^v$ be the MDP with transition $\mathbb{T}$ and mean reward function $R^v$, and $\mathcal{M} = \{M^v : v \in \Theta\}$ be the corresponding class of MDPs. We next show that $\mathcal{M}$ can be learned with polynomial process-based samples, but cannot be learned with polynomial outcome-based samples.

**Exponential Lower Bound for Outcome-Based Setting.**   When executing a policy $\pi$ in MDP $M^v$, we have $a_1 = \pi_1(s_1)$, $s_2 = a_1$, and $a_2 = \pi_2(s_2)$, and the data $(\tau_{\theta,\theta'}, R)$ observed are in the following form of trajectory together outcome-based rewards:

$$\tau_\pi = (s_1, a_1, s_2, a_2), \qquad R|\tau_\pi \sim \text{Bern}\left(\frac{1}{3}\text{ReLU}(\langle a_1, v \rangle - b) + \frac{2}{3}\right),$$

where $\tau_\pi$ is a deterministic function of $\pi$. In the following, we denote $a_\pi = \pi_1(s_1)$, and then $\mathbb{E}[R|\pi] = \frac{1}{4}\text{ReLU}(\langle a_\pi, v\rangle - b) + \frac{1}{2}$. Further, under $M^v$,

$$J(\pi) = \frac{2}{3} + \frac{\varepsilon}{3}\mathbf{1}\{a_\pi = v\},$$

and in particular, $J(\pi^\star) = \frac{2}{3} + \frac{\varepsilon}{3}$. Therefore, for any policy $\pi$, it is $(\varepsilon/3)$-optimal under $M^v$ only when $a_\pi = v$. Hence, we can apply the standard lower bound argument for multi-arm bandits (see e.g., Lattimore and Szepesvári, 2020) to show that: If there any $T$-round algorithm that returns an $(\varepsilon/3)$-optimal policy with probability at least $\frac{3}{4}$ for any MDP $M^v \in \mathcal{M}$, then it must hold that $T \geq c\frac{N}{\varepsilon^2}$ (where $c > 0$ is an absolute constant). Setting $\varepsilon = \frac{1}{3}$ completes the proof of the lower bound. $\qquad\square$

**Polynomial Upper Bound with Process-Based Samples.** Notice that for fixed $v \in \Theta$, under $M^v \in \mathcal{M}$, we have

$$J(\pi^\star) - J(\pi) = \frac{\varepsilon}{3}[1 - \mathbf{1}\{a_\pi = v\}] = \frac{1}{3}[\langle v, v\rangle - \langle a_\pi, v\rangle].$$

Thus, for any $\theta \in \Theta$, we define $\pi^\theta$ as $\pi_1^\theta(s) = \pi_2^\theta(s) = \theta$ for $\forall s \in \mathcal{S}$. Then it holds that under $M^v$,

$$\mathbb{E}\left[\frac{1}{3} - R_2 \,\middle|\, \pi^\theta\right] = \frac{1}{3}\langle \theta, v\rangle.$$

Therefore, given access to process reward feedback, we can reduce learning $\mathcal{M}$ to learning a class of linear bandits. Hence, for any $\alpha > 0$, with process reward feedback, there are algorithms that returns an $\alpha$-optimal policy with high probability, using $T \leq \widetilde{O}(d^2/\alpha^2)$ episodes with process rewards (see e.g., Dani et al., 2008). $\qquad\square$

