# OpenReview forum: "Outcome-Based Online Reinforcement Learning: Algorithms and Fundamental Limits"
_NeurIPS.cc/2025/Conference — NeurIPS 2025 poster_

### Official Review · Reviewer_YCwT · 2025-06-27

**Clarity:** 3
**Significance:** 3
**Originality:** 3
**Rating:** 4
**Confidence:** 4

**Summary:**

This paper presents a comprehensive analysis of outcome-based online reinforcement learning, addressing the fundamental credit assignment challenge when rewards are observed only at the end of a trajectory. The authors develop a provably sample-efficient algorithm (Algorithm 1) for settings with general function approximation, achieving a sample complexity of $\tilde{O}(C_{cov}H^{3}/\epsilon^{2})$. A key contribution is the identification of a fundamental exponential separation, revealing that for certain MDPs, outcome-based feedback is unavoidably less efficient than per-step rewards. For the special case of deterministic MDPs, the paper also proposes a significantly simpler and more computationally efficient variant (Algorithm 2) that eliminates the strong completeness assumption required by the primary algorithm. Finally, the framework is extended to preference-based feedback settings, demonstrating that similar statistical efficiency can be achieved.

**Questions:**

On the Computational Feasibility of Algorithm 1: Beyond the special case of deterministic MDPs, could you elaborate on potential pathways to make this optimization computationally feasible, even if it requires approximations? For example, are there specific structural assumptions on the function classes $\mathcal{F,R,G}$ (beyond determinism) or approximation techniques (e.g., alternating optimization, gradient-based methods for the outer loop) that you believe could make Algorithm 1 practical?

**Ethical Concerns:**

["NO or VERY MINOR ethics concerns only"]

**Final Justification:**

I would like to keep my original score since the authors have addressed my concern.

**Limitations:**

yes

**Quality:**

3

**Strengths And Weaknesses:**

- strengths:
	- Foundational and General Theoretical Framework: This paper provides what appears to be the first comprehensive theoretical analysis for online RL with outcome-based feedback under the challenging general function approximation setting. By developing a provably sample-efficient algorithm, the authors establish a rigorous and foundational contribution that is applicable to complex environments where tabular methods are insufficient.
	- Insight on Fundamental Limits: A key strength is the paper's lower bound analysis, which establishes a fundamental exponential separation between outcome-based and per-step feedback. It proves that for certain MDPs, outcome-based feedback is inherently less efficient, requiring exponentially more samples to learn a near-optimal policy. This result clearly defines the statistical barriers of the problem and provides crucial context for when outcome-based learning is tractable.
	- Practical Relevance and Extensions: The paper's impact is broadened by its thoughtful extensions and methodological insights. The approach is generalized to handle preference-based feedback, directly connecting the theory to practical Reinforcement Learning from Human Feedback (RLHF) scenarios. For deterministic MDPs, the authors provide a much simpler and more computationally efficient algorithm (Algorithm 2) that eliminates the need for a completeness assumption. Finally, the work offers clear guidance by demonstrating that jointly optimizing the reward and value functions is essential, as simpler decoupled approaches can fail.


- weaknesses
	- The central contribution, Algorithm 1, hinges on solving a joint optimization problem (Eq. 6) to estimate the value and reward functions. While theoretically elegant, this step poses significant practical challenges. The objective function itself involves a Bellman loss term, $L_D^{BE}(f;R)$, which requires computing an infimum over function class $\mathcal G$. This creates a nested, max-min-like optimization problem. In the general function approximation setting, where $\mathcal F$ and $\mathcal R$ are typically non-convex (e.g., neural networks), solving this outer maximization for a global optimum is generally intractable.

---

> ### Author Rebuttal · Authors · 2025-07-31
>
> We thank the reviewer for the review. Below, we address the concerns raised in the review:
>
> > The central contribution, Algorithm 1, hinges on solving a joint optimization problem (Eq. 6) to estimate the value and reward functions. While theoretically elegant, this step poses significant practical challenges. The objective function itself involves a Bellman loss term, which requires computing an infimum over function class . This creates a nested, max-min-like optimization problem. In the general function approximation setting, where  and  are typically non-convex (e.g., neural networks), solving this outer maximization for a global optimum is generally intractable.
>
> We admit that to solve the optimization problem in Algorithm 1 can be computationally intractable without proper assumptions. Broadly speaking, in the literature of RL theory, achieving computational efficiency RL with general function approximation has long been an open problem. To the best of our knowledge, it is only partly resolved under restricted problems (e.g., MDPs with linear structure) or stronger access oracle (reset).
>
> We note that under reset access, the max-min optimization in our algorithm can indeed be replaced by a maximization problem, following the approach of [A]. Achieving computational tractable outcome-based RL beyond the setting of reset access can be an important direction of future work.
>
> ---
>
> > On the Computational Feasibility of Algorithm 1: Beyond the special case of deterministic MDPs, could you elaborate on potential pathways to make this optimization computationally feasible, even if it requires approximations? For example, are there specific structural assumptions on the function classes  (beyond determinism) or approximation techniques (e.g., alternating optimization, gradient-based methods for the outer loop) that you believe could make Algorithm 1 practical?
>
> As mentioned above, with reset access, the algorithm can be modified to avoid the max-min optimization. In addition, gradient-based approximation/alternating optimization can indeed be applied to the max-min optimization problem, assuming that the function classes are parametrized (e.g., class of neural networks). Particularly, under certain regularity conditions, the existing convergence results may be applied, showing that a local minima can be found. We believe analyzing the optimization landscapes of the GOLF-based algorithms is an important future direction for RL theory.
>
> [A] The Power of Resets in Online Reinforcement Learning. Zakaria Mhammedi, Dylan J. Foster, Alexander Rakhlin.

---

> > ### Comment · Reviewer_YCwT · 2025-08-09
> >
> > Thank you for your response.

---

### Official Review · Reviewer_mCk3 · 2025-06-30

**Clarity:** 4
**Significance:** 3
**Originality:** 3
**Rating:** 5
**Confidence:** 3

**Summary:**

The paper studies outcome based (aka trajectory feedback, or aggregate feedback) RL in the general function approximation setup, subject to the standard coverability, realizability and completeness assumptions typically required for sample efficient learning with general function approximation.

In more detail, the algorithms in the paper receive as input a class of reward functions $\mathcal R$ and a class of Q functions $\mathcal F$, and output an $\epsilon$-optimal policy after roughly $C_{\rm cov}N/\epsilon^2$ interaction episodes, where $C_{\rm cov}$ denotes the coverability coefficient and $N$ the covering number of $\mathcal R$ and $\mathcal F$.
An additional error term scales linearly with the violation of the Bellman completeness and realizability assumptions.

The algorithms presented include:
1. The fundamental trajectory feedback setup, which is computationally inefficient in general.
2. A simplified algorithm for the case of deterministic dynamics, which is efficient assuming oracle access to $\mathcal F$.
3. An extension of the fundamental setup algorithm to the case of preference feedback.

Finally, a lower bound is established that demonstrates standard semi bandit feedback is easier than trajectory feedback in the general function approximation setup (in the sense that the former admits sample efficient algorithms, while the latter does not).

**Questions:**

* What is the reason for employing the reference policy in the exploration phase of Alg1 (similarly Alg3)? As you explained, you make no assumption regarding the distribution of actions it chooses. If this is the case, why not just stick with $\pi^{(t)}$ for the entire episode, in favor of simplicity?

**Ethical Concerns:**

["NO or VERY MINOR ethics concerns only"]

**Final Justification:**

I have decided to maintain my original rating favoring acceptance. Following additional discussions, it appears computational efficiency, while being a weakness of the proposed method, is not by any means a major one or very specific to this work.

As noted by the authors and in the AC / reviewers discussions, the issue of computational intractability is common in many algorithms in RL with general function approximation.

**Limitations:**

Yes

**Quality:**

3

**Strengths And Weaknesses:**

### Strengths
* This work is the first to study trajectory feedback in the general function approximation setup. Prior works only study the tabular and linear setups.
* Three algorithms of independent interest are studied.
* The paper is well written with clear presentation, and is easy to follow.

### Weaknesses
* Nothing stands out.

### Minor comments
- Line 39: "where the reward function is assumed to be well structured" --- some of the works mention consider the tabular setup, and do not make any particular assumptions on the reward shape. What exactly do you mean by "well structured" or "well behaved"? If I understand correctly the distinguishing feature of the present paper is that it obtains rates independent of $S$. But, this does not relate to well behavedness of the reward function. Works that study the linear setup such as Cassel et al. 24 consider linear MDPs, which is a much stronger assumption than just linear reward.
- Line 126: "Bellman operator" might be a better heading?
- Line 175: What is $d$?
- Line 256: "which is widely used on RLHF literature" (on $\to$ in)
- Line 300: "Is outcome-based samples are statistically equivalent"

---

> ### Author Rebuttal · Authors · 2025-07-31
>
> We thank the reviewer for the detailed review. Below, we address the concerns raised in the review:
>
> > **Line 39 Meaning of "well structured" or “well behaved” reward functions**
>
> We use this adjective to describe that the reward function is either linear with respect some features of state-action pairs, or slightly more generally, admitting bounded *trajectory* eluder dimension. Assumptions on well-behaved reward function is very common in previous work, including in tabular MDP (where the reward function is automatically linear with respect to the $|\mathcal{S}|\times |\mathcal{A}|$-dimensional feature) and linear MDP (where the linear reward function is part of the linearity structure).
>
> We will clarify this in the revision.
>
> ---
>
> > **Line 126 Bellman operator as the heading?**
>
> Thanks for the suggestions. We will update it in our revision.
>
> ---
>
> > **Line 175 what’s $d$?**
>
> In the setting where the function classes are parametric, $d$ is the dimension of the parametric model, i.e.. the number of “free parameters”. For example, when the function classes are linear with respect to a feature map, $d$ is the dimension of the feature.
>
> ---
>
> > **Line 256 and Line 300**
>
> Thanks for pointing out the typo, we will fix them in the version.
>
> ---
>
> > **Reason and choice of reference policy in Alg1 and Alg3**
>
> The point of adopting the policy $\pi\circ_h \pi_{\mathrm{ref}}$ is to make the algorithm more exploratory. The fixed reference policy serves as a comparator for the policy $\pi^{(t)}$, and it has to be the same across different iterations. Based on this reasoning (and our theoretical analysis), we cannot choose $\pi_{\mathrm{ref}}$ to be $\pi^{(t)}$, which changes over time.

---

> > ### Comment · Reviewer_mCk3 · 2025-08-09
> >
> > Thank you for the clarifications.
> >
> > After going through the rest of the reviews / rebuttals, I maintain my opinion that this is a good paper that offers a clear contribution to the theory of RL with trajectory feedback.

---

### Official Review · Reviewer_EJnZ · 2025-07-02

**Clarity:** 2
**Significance:** 3
**Originality:** 2
**Rating:** 4
**Confidence:** 3

**Summary:**

This paper studies online reinforcement learning with trajectory-wise reward feedback. While previous works have focused on structured rewards such as linear functions, this paper considers general function approximation. Under standard realizability and completeness assumptions, the authors propose an algorithm and establish an $\tilde{O}(H^3/\epsilon^2)$ sample complexity guarantee. They also provide theoretical analysis in the setting of preference-based learning. Finally, the authors present an example MDP in which all outcome-based reinforcement learning algorithms incur exponential sample complexity.

**Questions:**

1. Could the authors clarify whether the algorithm collects $H$ trajectories per iteration by executing the policy $\pi \circ h\,\pi_{ref}$?
2. In offline learning, $\mu$ usually refers to the data distribution. In the online setting, however, data is generated interactively over time. Could the authors clarify what $\mu$ specifically represents in online RL?
3. In the lower bound proof, the ReLU bandit example (Section 6.1.4) demonstrates a linear sample complexity in $T$. Is this because the example violates Assumption 1 and Assumption 2? Clarification on whether the lower bound arises from the failure of these assumptions would be appreciated.

**Ethical Concerns:**

["NO or VERY MINOR ethics concerns only"]

**Final Justification:**

My concerns have been addressed in the rebuttal phase.

**Limitations:**

Yes

**Quality:**

3

**Strengths And Weaknesses:**

### Strengths:
 1. This work is the first to provide a theoretical analysis of online reinforcement learning with outcome-based reward feedback under general function approximation.
 2. The authors present a counterexample MDP in which per-step reward RL algorithms are sample-efficient, whereas outcome-based algorithms suffer from exponential sample complexity. This highlights a fundamental difference between outcome-based MDPs and standard MDPs.

### Weakness
1. In the loop between steps 4 and 7 of the Algorithm 1, the policy $\pi \circ h\,\pi_{ref}$ is executed once per iteration to collect a trajectory. This means that each iteration yields $H$ trajectories in total, and the entire algorithm generates $TH$ trajectories over $T$ iterations. This differs from standard online RL algorithms, which usually collect only one trajectory per iteration. I haven't carefully checked the proof details, so it's unclear whether a typo error exists at this point. However, it seems that the theoretical guarantees rely on collecting $H$ times more trajectory data. If those additional trajectories were omitted, the sample complexity of the algorithm could increase accordingly.

2. Some notations are not explained, such as $d_h^{\pi}$ and $\mu_h$, although they are standard notations in offline learning.

3. In the definition of Bellman completeness, the Bellman operator is applied to vector-valued functions $f$ ($f_1,...,f_H$), whereas in Assumption 2, it is applied to scalar-valued functions $f_{h+1}$.

---

> ### Author Rebuttal · Authors · 2025-07-31
>
> We thank the reviewer for the detailed review. Below, we address the concerns raised in the review:
>
> > In the loop between steps 4 and 7 of the Algorithm 1, the policy $\pi\circ_h \pi_{\rm ref}$ is executed once per iteration to collect a trajectory. This means that each iteration yields $H$ trajectories in total, and the entire algorithm generates $TH$ trajectories over $T$ iterations. This differs from standard online RL algorithms, which usually collect only one trajectory per iteration.
>
> Indeed, our algorithm collects $H$ trajectories per iteration by executing the policy $\pi\circ_h \pi_{\rm ref}$ for each $h$. This strategy makes the algorithm actively explore the environment. While it is not a standard one in the literature of online RL, such a “per-step” exploration is commonly adopted in the literature on partially observable RL, where active exploration is crucial for sample efficiency guarantee.
>
> As a remark, our algorithm can also be modified so that it only collects one trajectory per iteration, by uniformly sampling a step h and executing $\pi\circ_h \pi_{\rm ref}$. Our analysis can similarly be adopted with such a modified algorithm. However, to keep the presentation as simple as possible, we chose to present the current version of the algorithm.
>
> ---
>
> > Some notations are not explained, such as $d^\pi_h$ and $\mu_h$, although they are standard notations in offline learning. …In offline learning, $\mu_h$ usually refers to the data distribution. In the online setting, however, data is generated interactively over time. Could the authors clarify what  specifically represents in online RL?
>
> We thank the reviewer for bringing up this subtle point. $d^\pi_h$ is defined to be the occupancy measure at layer $h$. More specifically, for any state $s\_h\in \mathcal{S}\_h$ and action $a_h$, $d^\pi\_h = \mathbf{Pr}\_{\tau\sim \pi}[(s_h, a_h)\in \tau]$, where $\tau$ is a trajectory sampled according to policy $\pi$. Additionally, $\mu_h$ appears in the definition of the coverability coefficient in online learning. The coverability coefficient is defined as
> $$
> C\_{\rm cov}=\min\_{\mu_1,\cdots,\mu_H}\max_{h\in [H],\pi\in\Pi}\left\|\left\|{\frac{d^\pi_h}{\mu_h}}\right\|\right\|\_{\infty},
> $$
> where there is an infimum over *all* distributions $\mu_1,\cdots,\mu_H$ over $\mathcal{S}\times\mathcal{A}$. By contrast, in offline RL, the sample complexity guarantees typically scale with the *concentrability coefficient*:
> $$
> C\_{\rm conc}=\max_{h\in [H],\pi\in\Pi}\left\|\left\|{\frac{d^\pi_h}{\mu_h}}\right\|\right\|\_{\infty},
> $$
> where $\mu\_1,\cdots,\mu\_H$ are the *fixed* offline data distribution. Therefore, the coverability coefficient equals the smallest possible concentrability among all possible offline data distributions. The above relation between the coverability coefficient and the concentrability coefficient provides a bridge between offline RL and online RL, and the coverability coefficient has been identified as a natural and intrinsic complexity of online RL (see e.g., (Xie et al., 2023), and also [A] and [B]).
>
> ---
>
> > In the definition of Bellman completeness, the Bellman operator is applied to vector-valued functions $f (f\_1,...,f\_H)$, whereas in Assumption 2, it is applied to scalar-valued functions $f_{h+1}$.
>
> Thank you for pointing out this typo. We will update the definitions in the revision, so that Bellman operator is defined on the vector-valued function and consistent with the existing work.
>
> ---
>
> > In the lower bound proof, the ReLU bandit example (Section 6.1.4) demonstrates a linear sample complexity in $T$. Is this because the example violates Assumption 1 and Assumption 2?
>
> In the ReLU bandit example, both Assumption 1 and 2 are satisfied, but the coverability $C_{\rm cov}=\exp(\Omega(d))$, as the set of all (possibly optimal) actions is exponentially large. We will clarify this in the revision.
>
> References:
>
> [A] The Power of Resets in Online Reinforcement Learning. Zakaria Mhammedi, Dylan J. Foster, Alexander Rakhlin.
>
> [B] Scalable online exploration via coverability. Philip Amortila, Dylan J Foster, Akshay Krishnamurthy.

---

> > ### Comment · Reviewer_EJnZ · 2025-08-02
> >
> > Thank you for the authors' reply. It was mentioned that the algorithm can be adapted to collect a single trajectory per iteration. I am wondering whether this modification would result in an increased sample complexity?

---

> > > ### Author Response · Authors · 2025-08-02
> > >
> > > Thank you for the follow-up question. We would like to clarify the following point: the sample complexity of Algorithm 1 is distinct from the number of iterations. The sample complexity is outlined in Line 181, and it remains unaffected regardless of whether we require a single trajectory per iteration.

---

> > > > ### Comment · Reviewer_EJnZ · 2025-08-04
> > > >
> > > > Thanks. My concern has been addressed. I will raise my evaluation.

---

### Official Review · Reviewer_T9uz · 2025-07-03

**Clarity:** 3
**Significance:** 3
**Originality:** 3
**Rating:** 5
**Confidence:** 4

**Summary:**

The authors tackle reinforcement learning with outcome-based feedback in the case of general function approximation. Under the assumptions of coverability and completeness, they show that an optimistic algorithm that models the unknown reward function and balances the estimated value of the Q-function, the TD error, and the reward model error can achieve a sample complexity of $T \geq O(C_{\text{cov}} N(1/T) H^2/\epsilon^2)$ episodes. A simplified algorithm is presented in the case of deterministic MDPs that only requires realizability instead of completeness. They extend the first algorithm to preference-based RL, and conclude by providing a hard instance, showing that learning from outcome-based feedback can be exponentially harder than process reward feedback when coverability is not satisfied but the low Bellman eluder dimension assumption is.

**Questions:**

1. If the assumption that the outcome-based feedback is given by the sum of rewards is relaxed, how do you think the results would change?
2. How does your paper compare to Kausik et al. (2024)?

I was going to award this paper a score of 5 (Accept), before noticing the concerns outlined above that affect the significance and originality of the paper. I will be very happy to increase my score if my concerns are addressed.

**Ethical Concerns:**

["NO or VERY MINOR ethics concerns only"]

**Final Justification:**

The author's rebuttal has satisfactorily addressed my concern previously highlighted at the end of the questions section. As promised, I have increased my score.

**Limitations:**

Yes.

**Quality:**

4

**Strengths And Weaknesses:**

## Strengths
1. **Strong theoretical contribution in a highly salient setting**.
- The paper addresses a highly salient setting with a thorough mathematical analysis, providing trajectory-only feedback theory with matching lower bounds for general function approximation. The proofs appear to be sound.
- The contributions are varied and thorough, consisting of a statistically efficient model-free algorithm, a simplified algorithm for deterministic MDPs, extension to preference-based feedback, and a lower bound that completes the picture -- while coverability is sufficient for RL with outcome-based feedback consisting of the sum of rewards, the low Bellman eluder dimension assumption is not.
2. **Original algorithms**.
- The authors provide two original algorithms for RL theory with general function approximation, that very refreshingly are not derivatives of GOLF. The authors also show that joint optimization of the reward model and Bellman loss is necessary as well.
- In particular, it is very nice that the authors adopt the MEX approach, instead of maintaining per-step confidence sets as in GOLF, to achieve sufficient optimism in Algorithm 1.
3. **Clear writing and presentation**.
- The paper is well-written and very easy to understand.

## Weaknesses
1. **Claim of first comprehensive analysis of outcome-based feedback in online RL with general function approximation**.
- Relevant work exists that the authors do not cite or compare their results to. Kausik et al. (2024) view cardinal (good/bad) and dueling (preferential) feedback in RLHF as partially-observed reward states, and provide guarantees for model-based (in the tabular setting) and model-free (a GOLF derivative in the general function approximation setting) RL. The latter uses a history-aware analogue of the Bellman-eluder dimension. In my opinion, the setting is highly similar, even though the approach may be different, and it may not be appropriate to say that this is the first comprehensive analysis of the problem in online RL with general function approximation.
2. **Outcome model structure**.
- Further, the claim that this gives a comprehensive analysis of RL with outcome-based feedback may not be appropriate as well. In my view, the contribution is not as broad as the paper claims for the below reason.
- The authors assume that the outcome-based feedback observed is given by the sum of the rewards in the trajectory. This is not necessarily the case in practice (for instance, one can easily imagine a threshold outcome that yields 1 if the sum of rewards is high enough and 0 otherwise), and is an assumption that other prior work (see Kausik et al. (2024) for an example) does not make.
- In my view, it is quite possible that this assumption is the reason why this paper achieves stronger results than Kausik et al. (2024) in the case where no intermediate feedback is observed in their setting. To be more specific, the fact that the outcome-based feedback is assumed to be the sum of the rewards in this paper may be the very reason why the credit assignment problem is solvable in this setting (under coverability) but not that of Kausik et al. (2024).
- However, this is not stated as an assumption, but is presented as part of the setting itself. I do not think this is good practice. I am of the opinion that this should be highlighted and presented as an assumption, due to the above remarks.
- This ultimately limits the scope of the paper, and one can conclude that this paper therefore may only tackle a special case of outcome-based feedback where the outcome is the sum of the rewards.

## References
1. Cheng et al. (2022), Adversarially Trained Actor Critic for Offline Reinforcement Learning
2. Kausik et al. (2024), A Theoretical Framework for Partially Observed Reward-States in RLHF

---

> ### Author Rebuttal · Authors · 2025-07-31
>
> We thank the reviewer for bringing the work of Kausik et al. (2024) to our attention. To facilitate a clear comparison, we outline the key distinctions between the two settings:
>
> - Our setting (A): The outcome feedback is (a noisy observation of) the sum $R_1(s_1,a_1)+\cdots+R_H(s_H,a_H)$
>
> - Setting of Kausik et al. (2024) (B): The outcome feedback is a (possibly sparse) vector $(R_h(s_h,u_h,a_h))_{h\in \mathcal{H}}$, where $u_h$ is the *unobserved* reward state that can depend on the history up to step $h$ in an arbitrary way.
>
> We would like to note that setting (A) represents the most commonly studied outcome-reward model in recent theoretical works (Efroni et al., 2021; Pacchiano et al., 2021; Chatterji et al., 2021; Wu and Sun, 2023; Wang et al., 2023; Cassel et al., 2024; Lancewicki and Mansour, 2025). When we refer to providing a "comprehensive theoretical analysis of outcome-based online RL," we are specifically addressing this well-established setting (A).
>
> We acknowledge that setting (B) presents greater learning challenges, as the reward structure may depend on the entire history. However, we would like to discuss some limitations when directly applying the results of Kausik et al. (2024) to our outcome feedback setting (A):
>
> - Trajectory space complexity: The upper bounds in Kausik et al. (2024) scale with variants of the eluder dimension over the trajectory space, which must be $\Omega(|\mathcal{A}|^H)$ in worst-case scenarios (e.g., combination locks). The conditions under which trajectory eluder dimension can be bounded by other relevant problem parameters remain largely unexplored. Notably, it cannot be bounded by the per-step eluder dimension, as demonstrated by our lower bound. While such dependence on trajectory eluder dimension may be unavoidable under the fully general setting (B) due to combination locks falling into this category, we believe there is value in exploring more refined analyses for specific subclasses.
>
> - One of our objectives is to avoid dependence on eluder dimension over trajectory space—an approach that aligns with recent advances in partially observable RL, which typically leverage specific structural properties (observability, decodability, stability, etc.) other than trajectory eluder dimension to obtain clearer bounds. We identify a broad and practically relevant class of MDPs that are learnable under setting (A), with upper bounds scaling with the coverability of the underlying MDP—a relatively intuitive and well-understood complexity measure.
>
> - Importantly, coverability serves as an intrinsic complexity measure of the MDP structure and remains bounded even when the eluder dimension of the value function class becomes unbounded (as with neural networks).
>
> While we recognize that setting (B) is indeed more general than (A), setting (A) remains significantly more challenging than learning with per-step (process) reward feedback. Previous works addressing setting (A) have primarily focused on linear rewards or functions with manageable trajectory eluder dimension. In this context, we believe our upper bounds using state-action coverability represent a meaningful theoretical contribution that involves several novel analytical techniques.
>
> We are grateful for the reviewer's observation that the phrase "comprehensive theoretical analysis of outcome-based online RL" could be clearer. In our revision, we will explicitly highlight that: (1) this work specifically addresses "sum of rewards" outcome feedback, and (2) we will provide appropriate context regarding other more general outcome feedback models considered in related literature.
>
> ---
>
> > If the assumption that the outcome-based feedback is given by the sum of rewards is relaxed, how do you think the results would change?
>
> When the outcome feedback takes the form of a general, non-linear function $R(s_1,a_1,\cdots,s_H,a_H)$, our upper bounds would indeed hold, but with coverability measured over the trajectory space rather than state-action space. However, trajectory coverability is typically much larger than standard coverability and, in worst-case scenarios, grows exponentially with the horizon—a highly undesirable property.
>
> This exponential dependence appears to be fundamental rather than an artifact of our analysis. For example, for the threshold outcome model (which outputs 1 if the sum of rewards exceeds a fixed threshold and 0 otherwise), it is possible to embed any H-step combination lock into an H-step MDP with a single state (as demonstrated by Kausik et al., (2024)). This construction shows that any algorithm operating with only threshold feedback must incur an exponential sample complexity $\Omega(|\mathcal{A}|^H)$ in the worst case.
>
> We hope this clarification helps position our work appropriately within the broader outcome-based RL literature, and we thank the reviewer again for this valuable feedback.

---

> ### Comment · Reviewer_T9uz · 2025-08-05
>
> I thank the authors for the clarification. It does indeed help position this paper within the broader outcome-based RL literature. I have no further questions, but will defer any finalized any score adjustments to after discussion with the other reviewers.

---

### Official Review · Reviewer_R6NT · 2025-07-03

**Clarity:** 2
**Significance:** 3
**Originality:** 3
**Rating:** 4
**Confidence:** 3

**Summary:**

The paper proposes an algorithm with sample complexity bound for outcome-based online RL. The algorithm can handle general function classes for approximation. The algorithmic framework is further simplified for deterministic MDPs and extended to preference-based RL to demonstrate the generic nature of the approach. Finally, lower bounds are derived for both the per-step reward-based and outcome-based RL settings. This shows exponential deviation among hardness of these two frameworks in some well-crafted MDPs.

**Questions:**

1. Can you provide some examples of $N(\alpha)$ and $N_T$ for some function classes? Present description in line 175 is vague to me.
2. The lower bounds are interesting for future research but no detail or intuition on the "hard" MDP construction is provided. Can you summarise why the specific two-layer MDP constructed exposes this gap? What is the bottleneck?
3. How does the covering number scales with dimensions and structures of policy class? It can go exponential in problem parameters, no? Where can we get a better polynomial type bounds on MDP parameters using this complexity measure? Also do we have to take the min over all base measures $\mu$? If yes, explain the intuition?
4. The formal proof for Theorem 3 of preference based optimization passes through proposition 22 and thus, proposition 8, which tries to control the Hellinger distance between the Bernoulli distributions over trajectories induced by different reward models. Why it has to be the Hellinger distance between them? Why not TV or any other metric? Can you explain this choice?
5. Please address the issues in weaknesses.

**Ethical Concerns:**

["NO or VERY MINOR ethics concerns only"]

**Final Justification:**

The paper addresses an interesting problem of outcome-based online RL. The discussions addresses most of my questions. I would really recommending adding the discussions to the paper. Also, discussing limitations like intractability of the algorithm should be discussed in detail to pave way for future works. Thus, I recommend a borderline accept .

**Limitations:**

Yes

**Quality:**

3

**Strengths And Weaknesses:**

Strengths:
1. The problem setup is interesting and timely.
2. The algorithm extends the well-studied fitted Q-iteration to this setting, which is nice approach to bridge the per-step reward and outcome-based RL setups.
3. The analysis of preference-based RL provides a motivating instantiation of the outcome-based RL in the context of modern RLHF applications.
4. The lower bounds demonstrating the different between per-step and outcome-based RL are insightful for future research.

Weakness:
1. Limitations of the analysis or the framework are not discussed.
2. Regret is never introduced in the paper. Also the notation $R$ is confusing with reward (e.g. Eq. 11, 13 etc.).
3. The description of the comparator class is almost non-existent. I understand that the idea comes from another work but explaining how to choose it exactly, specially in preference-based setting, is necessary for clarity.
4. The proposed lower bound on T to satisfy the error bounds in Eq 7, 10 and 14 seems intangible to me as both sides depend on T, and I do not see how the left side scales with T for some useful function classes. Can you explain this?

---

> ### Author Rebuttal · Authors · 2025-07-31
>
> We thank the reviewer for the review. Below, we address the concerns raised in the review:
>
> > Limitations of the analysis or the framework are not discussed.
>
> Thank you for pointing out the omission regarding the limitations of our analysis/framework. We will update them accordingly in the future version of our paper.
>
> ---
>
> > Regret is never introduced in the paper. Also the notation $R$ is confusing with reward (e.g. Eq. 11, 13 etc.).
>
> Thanks for mentioning this issue. We will add the following definition in our revision:
> “The regret across $T$ steps is defined as $\sum_{t=1}^T (V^\star - V^{\pi_t})$, where $V^\star$ is the optimal value function and $V^{\pi_t}$ is the value function of the policy chosen at time $t$.”
> We will also clarify the notation $R$.
>
> ---
>
> > The description of the comparator class is almost non-existent. I understand that the idea comes from another work but explaining how to choose it exactly, specially in preference-based setting, is necessary for clarity.
>
> The comparator class $\mathcal{G}$ for the Bellman completeness is indeed commonly considered in the literature of model-free RL. Typically, it can be chosen to be the value function class $\mathcal{F}$, as long as the class $\mathcal{F}$ is rich enough (so that it is close under the Bellman operator). When the class $\mathcal{F}$ consists of neural networks, we may expect the Bellman completeness holds with small error, as guaranteed by the approximation theory of neural networks.
>
> ---
>
> > Can you provide some examples of $N(\alpha)$ and $N_T$ for some function classes? Present description in line 175 is vague to me.
>
> For example, when $F$ is a parametric class of dimension $d$, e.g. $d$-dimensional linear function, we have $\log N(F, \alpha)\asymp d\log(1/\alpha)$. When $F$ is some non-parametric class, e.g. Lipschitz functions in dimension $d$, we have $\log N(F, \alpha)\asymp \alpha^{-d}$.
>
> ---
>
> > The proposed lower bound on T to satisfy the error bounds in Eq 7, 10 and 14 seems intangible to me as both sides depend on T, and I do not see how the left side scales with T for some useful function classes. Can you explain this?
>
> As discussed, for parametric function class, we have $N_T\asymp d\cdot \log T$, where $d$ is (roughly) the number of parameters. For such classes, all of these conditions in Eq 7, 10 and 14 with both sides depending on $T$ hold when $T\ge M * \log T^2$ for some parameter $M$ possibly depending on $\epsilon, \delta, H$ and $C_{\rm cov}$. Therefore, this condition is automatically satisfied when $T \ge M \cdot \log^2 M$. This condition can be used as a surrogate condition in Eq 7, 10 and 14. We will clarify this in the revision.
>
> ---
>
> > The lower bounds are interesting for future research but no detail or intuition on the "hard" MDP construction is provided. Can you summarise why the specific two-layer MDP constructed exposes this gap? What is the bottleneck?
>
> The main intuition of Theorem 4 (the lower bound) is that, without bounded coverability, the outcome reward may not be well-behaved even when the per-step (process) rewards are generalized linear functions. Technical, the lower bound follows from the following two facts:
>
> - ReLU bandit setting is exponentially hard to learn
>
> - In linear bandit setting, there are algorithms with $O(d\sqrt{T})$ regret
>
> So in our two-layered MDP construction, we let the full problem to be a ReLU bandit, meanwhile we encode a linear reward structure in the first layer. Hence with the per-step reward model, the learner has access to the linear structure in the first layer, hence can achieve sublinear regret, while with the outcome reward model, the learner must deal with the ReLU bandit, which is exponentially hard.
>
> ---
>
> > How does the covering number scales with dimensions and structures of policy class? It can go exponential in problem parameters, no? Where can we get a better polynomial type bounds on MDP parameters using this complexity measure? Also do we have to take the min over all base measures $\mu$? If yes, explain the intuition?
>
> $C_{\mathrm{cov}}(\Pi)$ is the denotes the coverability of the policy class, i.e. the coverage of policies in $\Pi$ among all possible offline distributions $\mu$, hence in the formulation we have to take min over all base measures. This notion is very natural for measuring the complexity of $\Pi$, as it notes the minimal efforts for learning under the covariant shift of all policies in $\Pi$. When the state space and action space are discrete, this notion has another formulation that might be more intuitive:
>
> $C_{\mathrm{cov}}(\Pi) = \max_{h} \sum_{s_h\in S_h, a_h\in A}\sup_{\pi\in \Pi} d^\pi(s_h, a_h).$
>
> Here we briefly some common MDPs with bounded coverability guarantee:
>
> - Tabular MDPs: $C_{\mathrm{cov}}(\Pi) \leq |S||A|$
>
> - Block MDPs: $C_{\mathrm{cov}}(\Pi) \leq |S||A|$, where $S$ is the latent state space
>
> - Linear/low-rank MDPs: $C_{\mathrm{cov}}(\Pi) \leq d|A|$, where $d$ is the feature dimension
>
> - Sparse low-rank MDPs: $C_{\mathrm{cov}}(\Pi) \leq k|A|$, where $k$ is the sparsity level.
>
> As shown by the above examples, the coverability coefficient scales polynomially with relevant problem parameters, instead of exponentially.
>
> ---
>
> > The formal proof for Theorem 3 of preference based optimization passes through proposition 22 and thus, proposition 8, which tries to control the Hellinger distance between the Bernoulli distributions over trajectories induced by different reward models. Why it has to be the Hellinger distance between them? Why not TV or any other metric? Can you explain this choice?
>
> The main purpose of Proposition 8 is to provide non-asymptotic guarantees for maximum likelihood estimates (MLE). We choose the Hellinger distance primarily to facilitate the analysis, as it offers technical convenience in theoretical derivations. As the TV distance can be upper bounded by the Hellinger distance, Proposition 8 still holds if the Hellinger distance is replaced by TV distance. Our choice of formulation aims to maintain generality, extending the applicability of the results beyond the immediate scope of this paper.

---

### Decision · Program_Chairs · 2025-09-17

**Decision:**

Accept (poster)

**Comment:**

The paper studies the sample complexity of RL problems with outcome/aggregated/trajectory feedback with general function approximation. Assuming realizability and completeness, the authors prove sample complexity bounds of $\tilde{O}(C_{cov}H^3/\epsilon^2)$ (where $C_{cov}$ is the coverability coefficient), via an intractable algorithm. They also suggest a simplified more tractable approach for deterministic MDPs, which only requires realizability, and extend the results to preference-based feedback. Finally, the authors show an exponential separation between per-step feedback and outcome feedback when coverability is not satisfied.

The reviewers agreed that this work is a well-rounded paper with several contributions, including a new statistically-efficient algorithm for RL with trajectory feedback and general function approximation, separation from the per-step feedback model, a simplified algorithm for deterministic MDPs, and more. The only potential weakness that was discussed between the reviewers was the intractability of the general approach. It was agreed upon that, in light of the previous literature in similar settings, this is only a minor weakness. The authors are still encouraged to explore whether the additional coverability assumption could be utilized to design computationally-efficient algorithms.